# Convective Characteristics and Formation Conditions in an Extreme Rainstorm on the Eastern Edge of the Tibetan Plateau

Yongren Chen [1,2,3,4,*] and Yueqing Li [1,2]

1   Institute of Plateau Meteorology, China Meteorological Administration, Chengdu 610072, China; yueqingli@163.com
2   Heavy Rain and Drought–Flood Disasters in Plateau and Basin Key Laboratory of Sichuan Province, Chengdu 610072, China
3   Meteorological Disaster Defense Technology Center of Sichuan Province, Chengdu 610072, China
4   Sichuan Provincial Meteorological Observatory, Chengdu 610072, China
*   Correspondence: yr20060004@163.com

**Abstract:** From 7 July to 11 July 2013, an extreme rainstorm occurred in the Sichuan Basin (SCB) of China, which is located at the eastern edge of the Tibetan Plateau, causing severe floods and huge economic losses. The rainstorm event was associated with mesoscale convection systems (MCSs). In this paper, we analyze the evolution characteristics and formation conditions of the MCSs, and the results show that: (1) the continuous activity of MCSs was a direct cause of the formation of extreme rainstorms. Under an "east high and west low" circulation mode, the MCSs formed a "cloud cluster wave train" phenomenon from the plateau to the basin; that is, the MCSs over the basin developed strongly in the process of the MCSs over the plateau area weakening. (2) The activities of MCSs over the rainstorm area was related to ascending branches of the two vertical circulations and topographic gravity wave. Under the influence of meridional vertical circulation, MCSs could move in the south–north direction in the western SCB, while under the influence of zonal circulation, it was difficult for MCSs to develop in the descending airflow east of 106°E. (3) In the mountainous area of the western part of the SCB, the gravity wave stress was obvious and its direction was opposite to the direction of the lower southeast warm–moist airflow. This configuration was able to form a drag effect in the low-level airflow, which was conducive to the convergence of the wind field and strengthening of the vertical ascending movement. These findings help in further understanding the effects of vertical circulation and terrain on MCSs and extreme rainstorms.

**Keywords:** extreme rainstorm; mesoscale convection systems (MCSs); vertical circulation

## 1. Introduction

China is a country that experiences many rainstorms [1–4]. Among them, the rainstorms that occur in the upstream, steep-terrain region of the Yangtze River are notoriously difficult to understand and forecast, and have long represented a difficult problem for meteorological research and operations relating to rainstorms in China [5,6]. The Sichuan Basin (SCB) in southwest China (Figure 1a) is on the leeward slope of the eastern side of the Tibetan Plateau, and is characterized by distinct geomorphology such as steep terrain, mountain basins, lakes, and rivers. Moreover, it is the main path for the warm and wet southwesterly flow at low latitudes arriving in the eastern part of China. In summer, influenced by the Tibetan Plateau weather systems (e.g., the Tibetan Plateau vortex [7–11], southwest China vortex [12–15] and western Pacific subtropical high (WPSH) [16], not only are there many wide-ranging and persistent heavy rainfall weather events, but also a large number of localized rainstorm processes occurring in the SCB. These abnormally strong precipitation events as well as the secondary disasters caused by such strong precipitation often lead to substantial socioeconomic damage and loss. Therefore, as a typical area

of China's rainstorm weather, the SCB is known within the country for its high level of incidence regarding torrential rain and flooding.

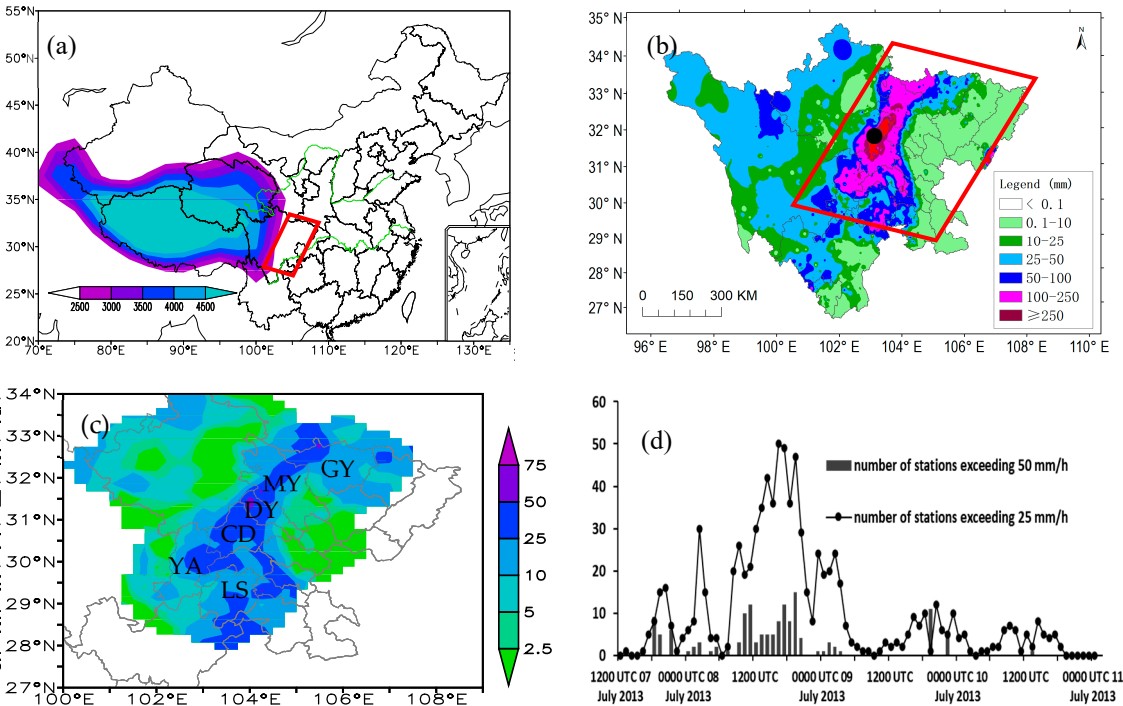

**Figure 1.** The location of the Sichuan Basin (**a**) where the shaded area is the elevation of the Tibetan Plateau (unit: m). The cumulative precipitation (**b**) (unit: mm) from 1200 UTC 7 July to 1200 UTC 11 July 2013. The maximum hourly precipitation (**c**) (unit: mm), and the evolution of AWS numbers recording ≥50 mm/h and 25 mm/h (**d**). In (**b**) the "●" denotes the location of Xingfu town, Dujiangyan. In (**c**), GY is Guangyuan, MY is Mianyang, DY is Deyang, CD is Chengdu, YA is Ya'an, and LS is Leshan.

Observational evidence has demonstrated robustly that the Tibetan Plateau vortex, shear lines [17], and Southwest China Vortex (SWCV) are the major weather systems responsible for rainstorms in Sichuan; additionally, they have an important influence on the strong precipitation that occurs in the vast areas downstream of the Tibetan Plateau. The considerable research effort in recent decades in this regard has led to some meaningful achievements such as on the generation sources of the Tibetan Plateau vortex and SWCV in summer [18–20], the causes of their formation [7,21–23], and their thermodynamic structure and effects on rainstorms [24–27]. Recently, with the application of high-resolution data, the structure and organization of mesoscale convective systems (MCSs) within these weather systems have been further revealed, indicating their importance in inducing the strong precipitation associated with such vortices [15,25–29].

In fact, the formation processes of a rainstorm are complex, for example, the large scale conditions, the quantity of water vapor, the strong upward motion, the level of instability in the atmospheric stratification, together with the effects of weather systems and topographies at multiple scales, all act in a synergistic way in the occurrence of severe rainstorm weather [30]. The MCSs are usually the direct systems of the associated rainstorms and can be organized in a variety of ways [25,31–33]. Accordingly, the spatial distribution, intensity, and duration of precipitation processes will differ, and past studies have shown that the organization of MCSs is an important factor affecting their ability to produce rainstorms, particularly the activities of MCSs with a long lifespan [34–36].

In the present study, an extreme rainstorm that occurred in the SCB during 7–11 July 2013 was chosen for analysis. Compared with historical observation records, the rainstorm process had some extreme characteristics such as high cumulative rainfall and strong hourly

precipitation. According to the cumulative rainfall from 1200 UTC 7 July to 1200 UTC 11 July, the number of weather stations recording ≥250 mm reached 184, and the maximum value was 1108.1 mm, which broke the historical record in Sichuan held since 1951. Furthermore, the rainfall caused a series of geological disasters including mountain torrents and debris flows, resulting in heavy casualties and economic losses of several billion dollars. From the perspective of trying to analyze the evolution of MCSs and understand why the rainstorm only occurred in the west of the SCB and lasted so long, we investigated the possible causes of this rainstorm process and sought to provide results that can act as a reference for similar rainstorm forecasts in the future.

The rest of this paper is organized as follows. Section 2 describes the data sources and methods; Section 3 analyzes the observed precipitation and the systems that directly induced it; Section 4 analyzes the evolution of clouds in the MCSs; Section 5 explains the conditions that were favorable for the MCS activities; and Section 6 concludes the study.

## 2. Data and Methods

The data used in this study were as follows:

(1) Observational data including sounding radio data at 0000 UTC and 1200 UTC, brightness temperature (Tb) data from the FY2D geostationary weather satellite provided by the National Satellite Meteorological Center, China Meteorological Administration, and ground-precipitation data of automatic weather stations (AWSs).

(2) Global final (FNL) analysis data from the National Centers for Environmental Prediction (NCEP), with a spatial resolution of $1° \times 1°$ and temporal resolution of 6 h (0000 UTC, 0600 UTC, 1200 UTC, and 1800 UTC each day; hereafter referred to as NCEP_FNL; http://rda.ucar.edu/datasets/ds083.2 (accessed on 12 March 2021)).

In order to discuss the environmental characteristics of MCS activity, we analyzed the main group of convective parameters including convective available potential energy (*CAPE*), convective inhibition (*CIN*), lifted index (*LI*), K-index (*KI*), and precipitable water (*PW*). The *CAPE* [37] is calculated as

$$CAPE = g \int_{z_{LFC}}^{z_{EL}} \frac{\delta T_v}{T_v} dz \qquad (1)$$

where $g$ is the gravitational acceleration constant, the subscript "LFC" denotes the level of free convection, and the subscript "EL" denotes the equilibrium level; $T_v$ is the virtual temperature; and $\delta T_v = (T_v)_{parcel} - (T_v)_{env.}$. The convective inhibition equals the negative work done by the mean atmospheric boundary layer parcel as it rises through the stable layer to its level of free convection [38] and is calculated as

$$CIN = g \int \frac{T_{env.} - T_{parcel}}{T_B} dz \qquad (2)$$

where $T$ is temperature, and $T_B$ is an average temperature for the layer. According to Galway [39], George [40], and DeRubertis [41], *LI* expresses the temperature difference between a lifted parcel and the surrounding air at 500 mb, and is negative when the parcel is warmer than its environment.

$$LI = (T_{500-hPa})_{env.} - (T_{500-hPa})_{parcel} \qquad (3)$$

For the *KI*, it serves as a predictor for thunderstorms that produce heavy rain and possibly flash flooding and its formulation is

$$KI = Td_{850} + (T_{850} - T_{500}) - DD_{700} \qquad (4)$$

Here, $Td_{850}$ is dewpoint at 850 hPa, and $DD_{700}$ is dewpoint depression. The $PW$ is defined as the depth of liquid water, which is the total mass of water vapor in a column with unit cross-sectional area [42]:

$$PW = \int_0^\infty q\rho dz \tag{5}$$

where $q$ is the specific humidity and $\rho$ is the air density. $PW$ has units of kilograms per meter squared, but it is often expressed in millimeters of equivalent water depth.

To analyze the dynamic and thermal conditions of rainstorm occurrence, we calculated the physical quantities such as pseudo-equivalent potential temperature ($\theta_{se}$) and vertical helicity. $\theta_{se}$ is an important parameter in MCSs analysis, according to Davies-Jones [43], and $\theta_{se}$ can be calculated as

$$\theta_{se} = T(\frac{1000}{p-e})^{R_d/C_{pd}} \exp(L \cdot r/C_{pd}T_c) \tag{6}$$

where $r = 0.6222^{e/(p-e)}$ is the water vapor mixing ratio; $T_c$ is the temperature at condensation height; $e$ is the vapor pressure; $p$ is the pressure; $T$ is the temperature; $L = (2.501 - 0.002370t) \times 10^6$ J·kg$^{-1}$; and $R_d/C_{pd} = 0.2854$. Therefore, $\theta_{se}$ is a comprehensive physical quantity that includes temperature, humidity, and pressure, and can reflect the varying characteristics of both moisture and temperature.

The helicity is a physical quantity that characterizes the dynamic characteristics of the fluid when it rotates and moves in a corkscrew fashion in the direction of the axis of rotation. It is often used as a diagnostic for rotating supercell thunderstorms [44] and defined as [44,45]

$$H = \vec{V} \bullet (\nabla \times \vec{V}) = \underbrace{u(\frac{\partial w}{\partial y} - \frac{\partial v}{\partial z})}_{(a)} + \underbrace{v(\frac{\partial u}{\partial z} - \frac{\partial w}{\partial x})}_{(b)} + \underbrace{w(\frac{\partial v}{\partial x} - \frac{\partial u}{\partial y})}_{(c)} \tag{7}$$

where $\vec{V}$ is the windspeed vector and $\vec{V} = u\vec{i} + v\vec{j} + w\vec{k}$; $\nabla \times \vec{V}$ is the three-dimensional vorticity; (a) is $x$ component of helicity, (b) is $y$ component of helicity, and (c) is $z$ component of helicity (also called vertical helicity, $VH$):

$$VH = w(\frac{\partial v}{\partial x} - \frac{\partial u}{\partial y}) \cong -\frac{1}{\rho g}\omega(\frac{\partial v}{\partial x} - \frac{\partial u}{\partial y}) \tag{8}$$

Here, $\omega$ is the vertical velocity of the $p$-coordinate system; $w$ is the vertical velocity of the $z$-coordinate system; $\rho$ is atmospheric density; and $g$ is the gravitational acceleration constant. The vertical helicity is a dynamic parameter that can reflect the maintenance status, the development, and the severity of weather systems [46,47].

In addition, as the rainstorm occurred near the terrain, the effect of surface gravity waves stress was further discussed. According to Xu et al. [48] and Xu [49], the surface gravity wave stress (also called surface momentum flux of gravity wave) is defined as

$$\vec{F}_0 = 8\bar{\rho}N\pi^2 \left|\vec{V}_0\right| \int_0^\infty \int_0^\pi (\cos\varphi, \sin\varphi)\cos(\varphi - \psi_0)\sqrt{1 - \frac{\cos^2(\chi_0 - \varphi)}{4Ri}} \left|\widehat{h}(K,\varphi)\right|^2 K^2 dKd\varphi \tag{9}$$

where $\bar{\rho}$ is the constant reference density; $N^2 = gd(\ln\theta_0)/dz$ is the Brunt–Väisälä frequency squared; $\vec{V}_0 = (U_0, V_0)$ is the base-state wind at the surface (height z = 0); $\varphi$ is the azimuth of base-state wind at the surface; $Ri \equiv N^2/\left|\vec{V}_z^2\right|$ is the ambient flow Richardson number and is greater than 0.25 for stably stratified flows; $\psi_0$ and $\chi_0$ represent the direction of $\vec{V}_0$

and $\vec{V}_z$, respectively; $\hat{h}$ is the 2D-Fourier transform of terrain height; and $\vec{K} = (k, l)$ is the horizontal wave vector, $K = \left|\vec{K}\right|$.

## 3. Observed Precipitation and Its Direct Inducing Systems

Extreme precipitation can be defined by the threshold method, that is, when precipitation exceeds a threshold, which can be a certain percentile of the total samples, the precipitation is called an extreme. For example, Luo et al. [50] defined the precipitation corresponding to the 99.9th percentile of each station as an extreme hourly precipitation based on historical observation data; Zhang and Ma [51] used a climatic statistical method to obtain a precipitation center of 230–260 mm/d, which can be used as an evaluation threshold of extreme precipitation events in the SCB; Chen and Li [52] analyzed the variation of precipitation based on AWSs and weather radar to show that strong hourly precipitation in the SCB varied from 20 to 80 mm/h, wherein precipitation above 50 mm/h is less frequent and can be considered as an extreme threshold. Based on these studies, in this paper, hourly rainfall exceeding 50 mm and daily precipitation exceeding 250 mm were used to classify extreme precipitation events.

For the rainstorms in the western SCB during 7–11 July 2013, the duration lasted up to four days, the number of AWSs recording 400–999.9 mm (Figure 1a) exceeded 100, and the number of stations recording ≥250 mm reached 184, with a maximum value of 1108.1 mm, which occurred at Xingfu town, Dujiangyan. From the evolution of the number of AWSs recording ≥25 mm/h and ≥50 mm/h (Figure 1b), the precipitation efficiency of the whole process was relatively high. Among them, the precipitation rate of ≥25 mm/h ran throughout the whole process and that of ≥50 mm/h was mainly concentrated in two stages (1200 UTC 7 July to 0600 UTC 8 July (the first period) and 0700 UTC 8 July to 0600 UTC 9 July (the second period)). These rainfall rates showed extreme characteristics, with the range of the second period being larger and the duration longer. Extreme hourly precipitation was mainly concentrated in the western SCB (Figure 1c). From 0600 UTC 9 July, the hourly precipitation weakened, but it continued to rain in the west of the basin. As of 1200 UTC, the cumulative maximum rainfall in this period (called the third period) exceeded 400 mm. Therefore, it was a significant extreme feature where the hourly precipitation reached rainstorm level and lasted for more than 12 h in the rainstorm. The entire precipitation process featured large cumulative rainfall and the total rainfall of one station exceeded 1000 mm, breaking the historical observation record, which was also a significant extreme feature.

According to the distribution of precipitation areas in each period (Figure 2a–c), in the first period, the rainstorm area was located in the west of the SCB (Ya'an, Guangyuan and Mianyang), the precipitation center was located at Mingshan Station of Ya'an (226.7 mm), the number of AWSs recording 50–99.9 mm was 83, and the number recording 100–249.9 mm was 47 (Figure 2a). In the second period, the rainstorm area expanded to the west of Chengdu, Deyang, and Mianyang, among which there were 60 stations that recorded precipitation values of ≥250 mm, 184 that recorded values of 100–249.9 mm, and 294 that recorded values of 50–99.9 mm (Figure 2b). The maximum value in this period was 741.3 mm, making it the strongest precipitation period. In the third period, hourly precipitation of ≥20 mm weakened; however, due to rainfall lasting for more than 30 h, the cumulative precipitation was still very high—there were 70 stations that recorded values exceeding 250 mm, 226 with values from 100–249.9 mm, and the maximum precipitation was 427.4 mm. Clearly, regardless of whether we focused on the hourly precipitation or the cumulative precipitation, this was certainly an extreme rainstorm event.

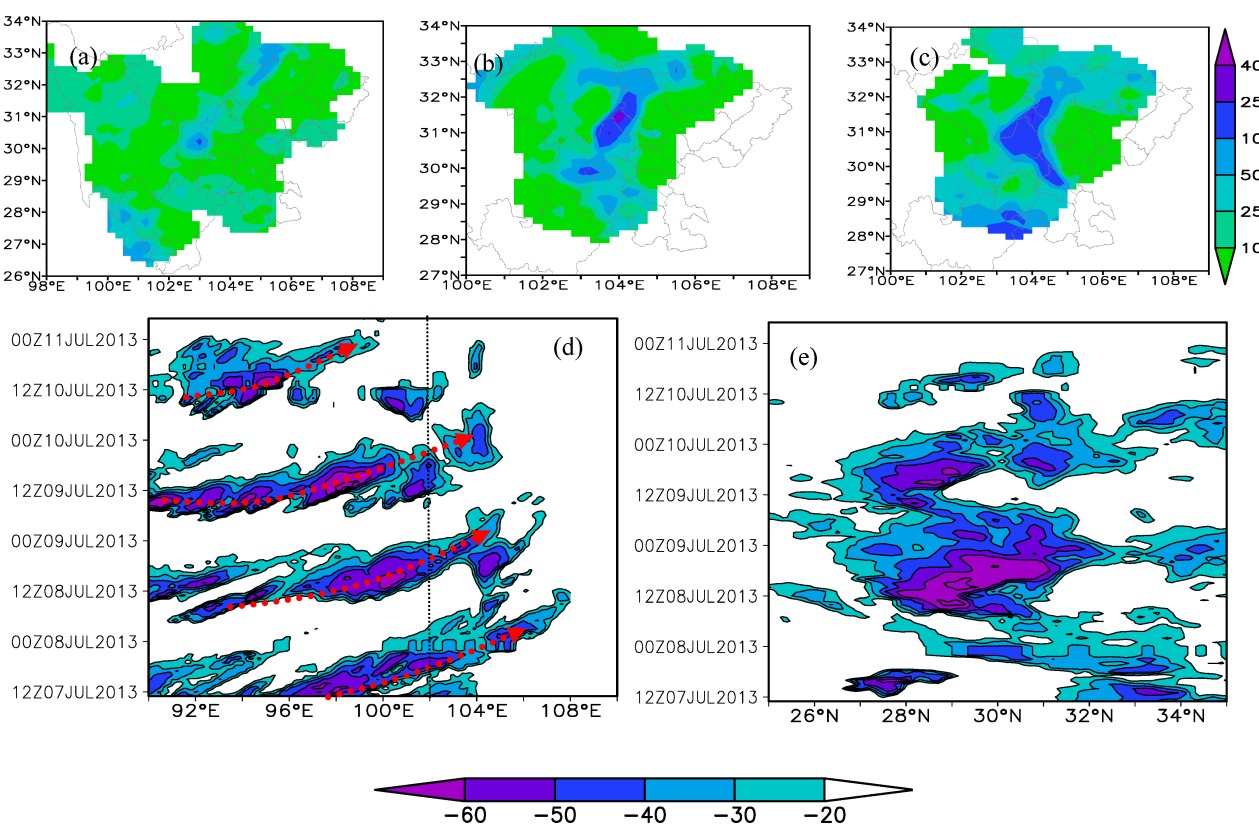

**Figure 2.** The cumulative precipitation of the first period (**a**) (1200 UTC 7 July to 0600 UTC 8 July), second period (**b**) (0700 UTC 8 July to 0600 UTC 9 July), and third period (**c**) (0700 UTC 9 July to 0000 UTC 11 July) (unit: mm), and cross-sections of hourly Tb (unit: °C) along the average of 31.5–32° N (**d**) and 103–103.8° E (**e**) (1200 UTC 7 July to 0000 UTC 11 July). In Figure 2d, the red arrow is the evolution direction of the convective cloud belt; the black dotted line is the boundary between the plateau and the basin.

According to the Tb evolution (Figure 2d,e), the three periods of heavy precipitation corresponded to three obvious activities of MCSs. From the plateau to the west of the SCB, three periods of MCS activity were observed; that is, between 1200 UTC 7 July and 0600 UTC 8 July, between 0700 UTC 8 July and 0600 UTC 9 July, and between 0700 UTC 9 July and 0600 UTC 10 July. After 0600 UTC 10 July, another period of MCS activity was also observed over the plateau area, but did not move to the west of the SCB—the precipitation in the basin was still affected by the weakened MCSs of the third period and the rainfall continued until 0000 UTC 11 July. Therefore, the rainstorm was closely related to frequent MCS activities, which moved to the rainstorm area and induced precipitation in the same area for about four days.

## 4. Cloud Evolution of the Mesoscale Convection Systems (MCSs)

To reflect the MCS activities of the three stages, we further analyzed the spatial evolution of Tb every 2 h (Figure 3). Figure 3a1–a8, b1–b14, c1–c16 reflect the MCS activities in stage 1, 2, and 3, respectively. In the first stage, rainstorm occurrence was mainly related to two MCSs—namely, MCS$_A$ and MCS$_B$. At 1400 UTC 7 July (Figure 3a1), MCSs had been active over the plateau, and the area of cold cloud (Tb < −50 °C) increased, but there were no MCSs over the west of the SCB at this time. In the subsequent evolution, the cold-cloud area of MCSs over the plateau decreased and showed a weakening trend; whilst at the same time, a new system (MCS$_A$) was generated over the west of the SCB, which was located downstream of the plateau. By 1800 UTC 7 July (Figure 3a3), MCS$_A$ was located over Ya'an, and then moved northwest of the SCB and lasted until 0400 UTC 8 July (Figure 3a8), which was a system that directly caused the strong precipitation of the

first stage. From 0400 UTC 8 July, in a weakened and eastward-moving convective cloud belt, another convective system again developed over Ya'an (MCS$_B$), but it lasted for a relatively short time of about three to four hours. Due to the successive influence of MCS$_A$ and MCS$_B$, Ya'an became the center of the rainstorm at this stage. In addition, both MCSs were generated at the front of the movement of the plateau convective cloud belt. When the MCSs over the plateau showed weakening, the MCSs over the SCB showed development. The evolution of MCSs over the plateau and basin showed an upstream and downstream effect. In the second stage, the weakened convective cloud that moved from the east of the plateau to the west of the SCB developed again and formed new MCSs. At 0800 UTC 8 July (Figure 3b2), MCS$_C$ and MCS$_D$ were generated over the west of the basin; meanwhile, new MCSs were also generated over the plateau. By 1000 UTC 8 July (Figure 3b3), the MCSs over the plateau and the west of the SCB had further developed, and the area of cold cloud with Tb < −50 °C continued to expand and lasted until 1400 UTC 8 July (Figure 3b5). From 1600 UTC 8 July (Figure 3b6), the MCS over the plateau began to weaken due to the lasting expansion of the cold-cloud area of MCS$_C$ and MCS$_D$ over the basin, which thus merged to form a larger system (MCS$_E$) that lasted until 0800 UTC 9 July (Figure 3b14), for about 16 h, which was the key convective system for the second-stage rainstorm. In this stage, with the weakened cloud over the plateau moving eastward, new convective systems over the SCB were generated, merged, and maintained for a long time, which was the main feature of convection and caused more severe precipitation. In the third stage, there were two periods of convection activity over the plateau area. The first started from 0600 UTC 9 July (Figure 3b13), in which MCSs developed over the plateau. By 1000 UTC 9 July (Figure 3c1), the cold-cloud area had expanded and entered a vigorous period, before then weakening and lasting until 0200 UTC 10 July. In this stage, the weakened cloud over the basin strengthened again, and a new convective system (MCS$_F$) was generated, which lasted until 2000 UTC 9 July (Figure 3c6). Another system (MCS$_G$) was generated at 1800 UTC 9 July (Figure 3c5) and lasted until 2200 UTC 9 July (Figure 3c7). The second period of convection over the plateau started at 0800 UTC 10 (Figure 3c12) and continued until the end of the rainstorm, but the MCSs did not move eastward to affect the rainstorm area. At this stage, the precipitation was caused by the weakened MCS in the previous stage. The corresponding precipitation intensity was weak, but due to the long duration of precipitation, the cumulative precipitation was also high.

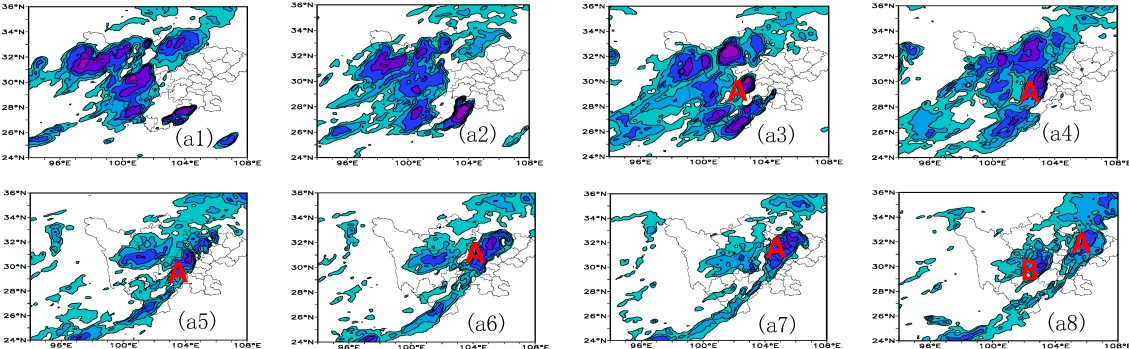

**Figure 3.** *Cont.*

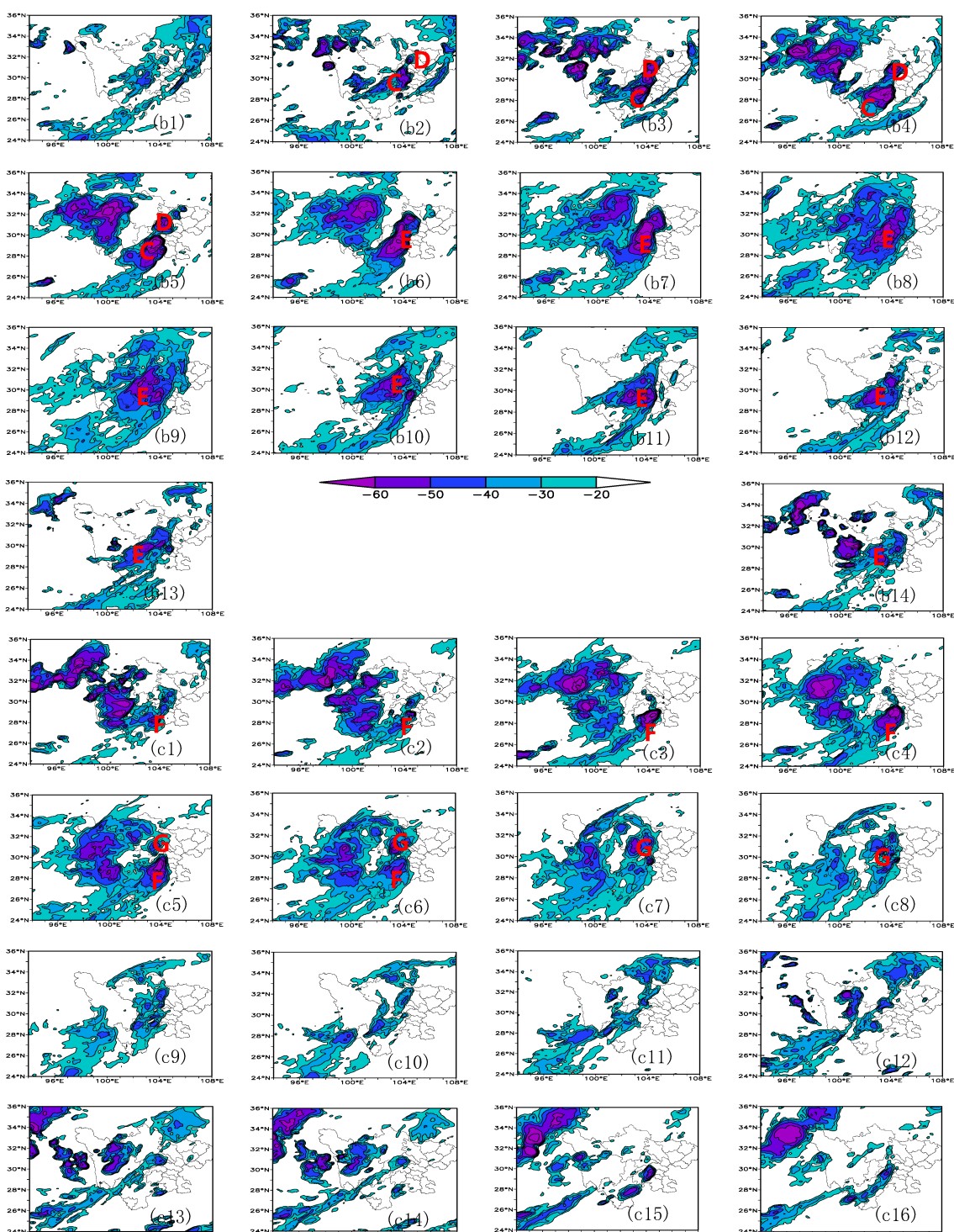

**Figure 3.** The Tb distribution of the first (**a1**–**a8**), second (**b1**–**b14**) and third period (**c1**–**c16**) in intervals of 2 h: (**a1**) 1400 UTC 7 July; (**a2**) 1600 UTC 7 July; (**a3**) 1800 UTC 7 July; (**a4**) 2000 UTC 7 July; (**a5**) 2200 UTC 7 July; (**a6**) 0000 UTC 8 July; (**a7**) 0200 UTC 8 July; (**a8**) 0400 UTC 8 July; (**b1**) 0600 UTC 8 July; (**b2**) 0800 UTC 8 July; (**b3**) 1000 UTC 8 July; (**b4**) 1200 UTC 8 July; (**b5**) 1400 UTC 8 July; (**b6**) 1600 UTC 8 July; (**b7**) 1800 UTC 8 July; (**b8**) 2000 UTC 8 July; (**b9**) 2200 UTC 8 July; (**b10**) 0000 UTC 9 July; (**b11**) 0200 UTC 9 July; (**b12**) 0400 UTC 9 July; (**b13**) 0600 UTC 9 July; (**b14**) 0800 UTC 9 July; (**c1**) 1000 UTC 9 July; (**c2**) 1200 UTC 9 July; (**c3**) 1400 UTC 9 July; (**c4**) 1600 UTC 9 July; (**c5**) 1800 UTC 9 July; (**c6**) 2000 UTC 9 July; (**c7**) 2200 UTC 9 July; (**c8**) 0000 UTC 10 July; (**c9**) 0200 UTC 10 July; (**c10**) 0400 UTC 10 July; (**c11**) 0600 UTC 10 July; (**c12**) 0800 UTC 10 July; (**c13**) 1000 UTC 10 July; (**c14**) 1200 UTC 10 July; (**c15**) 1400 UTC 10 July; (**c16**) 1600 UTC 10 July.

Obviously, on the front side of the weakened cloud belt over the plateau moving eastward, the newly developed MCSs were the precipitation systems directly responsible for the rainstorm. The size and duration of the cold-cloud area (Tb < −50 °C) of the MCSs affected the range and accumulation of heavy precipitation. The staged activity of the MCSs was the direct cause of the three rainstorm stages due to the frequent activities and long duration of MCSs, which caused extreme rainstorms. Moreover, the evolution of the convective cloud over the plateau and the basin formed an upstream–downstream effect: when the convective cloud over the plateau weakened, the convection over the basin showed a developing trend. The new and old MCSs formed a "cloud cluster wave train" phenomenon. To a large extent, the severity and duration of the MCS development determined the formation of extreme rainstorms; in other words, the development condition of the MCSs was the physical mechanism underpinning the heavy precipitation. However, what were the weather conditions conducive to the development of MCSs that stagnated over the rainstorm area? In this respect, the reason why MCSs were only active in the region west of 106° E is analyzed further below.

## 5. Favorable Conditions for MCS Activities

Past studies have shown that there are some conditions that affect the activities and development of MCSs, particularly environmental factors [53–55]. In the present study, to understand the initial conditions of MCSs over the steep terrain of the region, we analyzed the structure of the atmospheric circulation, the main convective parameters, and the gravity wave stress near this steep terrain.

### 5.1. Horizontal and Vertical Features of the Atmospheric Circulation

The occurrence of a rainstorm is often related to the interactions among weather systems of different scale; and only under the stable control of large-scale weather systems such as a subtropical high [56], long-wave trough, shear line, or large-scale low-pressure system can direct precipitation systems appear and produce strong precipitation. Using the sounding data to analyze the circulation of the MCS activities in the middle and lower levels (Figures 4 and 5), the sounding weather maps at 1200 UTC 7 July and 0000 UTC 8 July were selected to reflect the night and day circulation of the first stage; the weather maps at 1200 UTC 8 July and 0000 UTC 9 July were selected to reflect the second stage; and the weather maps at 1200 UTC 9 July and 0000 UTC 10 July were selected to reflect the third stage.

From the evolution of the 500-hPa circulation (Figure 4), the WPSH over the east side of the rainstorm area was less mobile, and there was a configuration of low-pressure weather systems (troughs, shears, etc.) over the plateau, forming an "east high and west low" circulation mode, which was the main weather background of the three stages of MCS activity. Under the influence of this circulation model, Tb showed that the MCSs of Tb < −50 °C were situated near and in front of the trough, but the range of MCS activity barely extended further than 106° E, possibly in relation to the circulation of the WPSH stagnating near 108° E. For example, at 1200 UTC 7 July in the first stage, there was a convective cloud belt composed of some MCSs at 90–102° E and located in the southwest wind flow at the bottom of the 500-hPa trough. By 0000 UTC 8 July, the trough had moved over the SCB, and correspondingly, the area of MCSs (Tb < −50 °C) over the western Sichuan plateau shrank significantly and the area of MCSs over the west of the basin expanded. This shows that, during the eastward movement of the trough, convective activity over the plateau weakened and convection over the west of the SCB enhanced. In the second stage, the "east high and west low" circulation mode was still maintained, and MCS activities showed similar characteristics as during the first stage. At 1200 UTC 8 July, there was a Tibetan Plateau low vortex (Yu et al. 2016) over (96° E, 34° N), and the accompanying shear line extended from the center of the low vortex to the western Sichuan Plateau. Near the shear line, a convective cloud belt formed again and followed several MCSs with Tb < −50 °C. By 0000 UTC 9 July (Figure 4c,d), convective activity had

weakened over the western Sichuan, and MCSs still existed over the west of the SCB. In the third stage, under the "east high west low" circulation background, a convective cloud belt appeared again over the western Sichuan Plateau at 1200 UTC 9 July, but weakened with a trough moving eastward. At 0000 UTC 10 July, a trough moved to the basin, convective activity weakened in the western Sichuan Plateau, and convection remained in the western basin. Clearly, the MCS activities in the three stages were basically located near the trough, but the MCSs over the western Sichuan Plateau weakened at night while those over the basin strengthened. This reflects the obvious characteristics of daily variation.

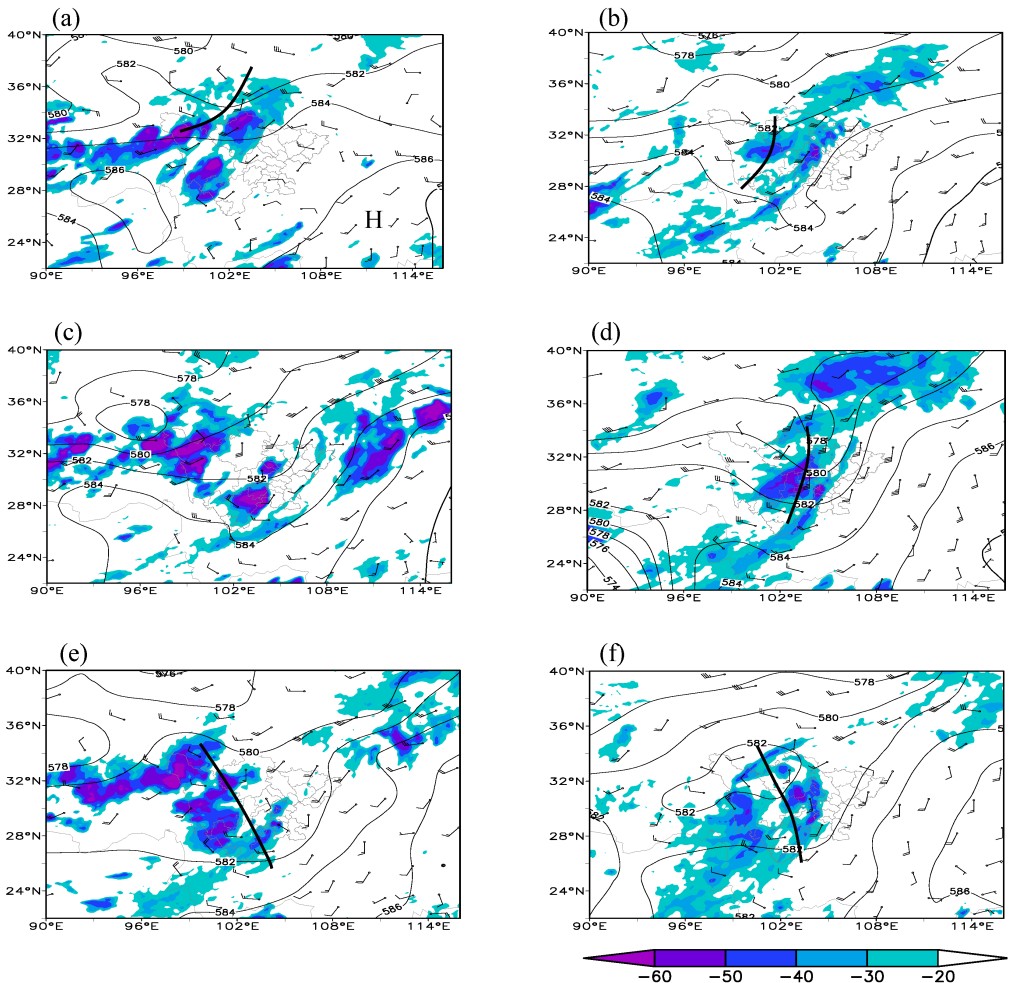

**Figure 4.** The 500-hPa geopotential height field (gpdam), wind field (m/s), and Tb (shaded: °C) at (**a**) 1200 UTC 7 July, (**b**) 0000 UTC 8 July, (**c**) 1200 UTC 8 July, (**d**) 0000 UTC 9 July, (**e**) 1200 UTC 9 July, and (**f**) 0000 UTC 10 July. H: WPSH, the thick solid line: trough.

For 700 hPa (Figure 5), there was a cyclonic vortex over the plateau, but its geopotential height was slightly higher during the day than at night. For example, in the first stage, the central geopotential height of the vortex reached 302 gpdam at 1200 UTC 7 July, and 304 gpdam at 0000 UTC 8 July; while in the second stage, there were values of 296 gpdam at 1200 UTC 8 July and 302 gpdam 0000 UTC 9 July; and in the third stage, 298 gpdam at 1200 UTC 9 July and 304 gpdam at 0000 UTC 10 July. In the west of the basin, the southerly wind showed an increasing trend at night such as the wind of WJ sounding station near the rainstorm center reaching 4 m/s at 1200 UTC 7 July and 8 m/s at 0000 UTC 9 July in the first stage. In the second stage, the 700-hPa wind reached 2 m/s at 1200 UTC 8 July and 10 m/s at 0000 UTC 9 July. In the third stage, it reached 4 m/s at 1200 UTC 9 July and 4 m/s at 0000 UTC 10 July. The development of MCSs over the west of the SCB may also

be related to the enhancement of low-level southerly airflow. Therefore, the persistence of the cyclonic circulation over the western Sichuan plateau was not only conducive to the frequent generation of plateau convection systems, but also affected the variation of MCS intensity over the western Sichuan Plateau. That is, when the geopotential height was descending, MCSs developed and the cold-cloud area of Tb < −50 °C expanded; and when the geopotential height was ascending, MCSs weakened or even disappeared. The development of the MCSs over the SCB was not only related to the 500-hPa trough, but also to the enhancement of low-level southerly airflow. Under the stable circulation situation of 500 hPa (the "east high and west low"), MCSs were active in the trough and shear zone. The MCSs over the plateau area weakened at night, while they strengthened over the basin at night, which formed a "cloud cluster wave train" phenomenon from the plateau to the basin.

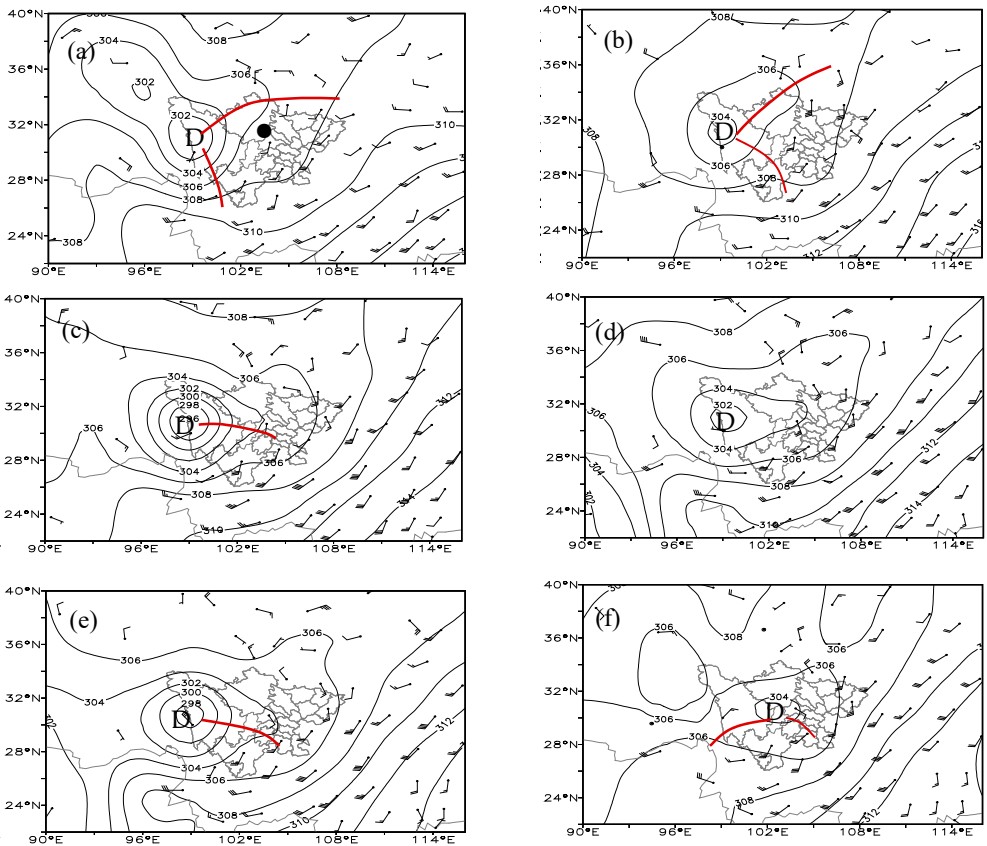

**Figure 5.** The 700-hPa geopotential height field (gpdam) and wind field (m/s) at (**a**) 1200 UTC 7 July, (**b**) 0000 UTC 8 July, (**c**) 1200 UTC 8 July, (**d**) 0000 UTC 9 July, (**e**) 1200 UTC 9 July, and (**f**) 0000 UTC 10 July. D: the cyclonic vortex, the thick brown line: shear line.

For the vertical structure of the airflow, due to the sparse availability of conventional detection data and the difficulty in analyzing such data in the rainstorm area, we further used the NCEP_FNL data for this part of the analysis. Figure 6 shows the vertical profiles of the vertical circulation and vertical velocity along 32° N and 105° E to reflect the characteristics of the vertical airflow in the rainstorm area. Among them, Figure 6a1–c1,a2–c2 reflects the meridional and zonal vertical circulation of the first stage, respectively; Figure 6d1–f1,d2–f2 reflects the second stage; and Figure 6g1–i1,g2–i2 reflects the third stage. It can be seen that the meridional profile showed a cyclonic vertical circulation between 500 and 300 hPa. In the zonal direction, there was an anticyclonic vertical circulation from 600 hPa to the ground, and the ascending branches of the two vertical circulations were located over the west of the SCB, which provided favorable dynamic conditions for convective activities. For example, in the first stage, at 1200 UTC 7 July, there was a cyclonic vertical circulation that appeared

at 29° N, which was located at 500 hPa, and its ascending branch was located in the area of 30–34° N. In the zonal direction, an anticyclonic vertical circulation was located at 600 hPa and below, the ascending branch of which was located in the area of 103–106° E, and east of 106° E was the descending branch of the airflow. The establishment of this airflow structure shows that the dynamic conditions over the rainstorm area had been more favorable in the early stage of the development of MCSs. By 1800 UTC 7 July, the structure of the cyclonic vertical circulation was more obvious and anticyclonic vertical circulation was maintained over the west of the SCB.

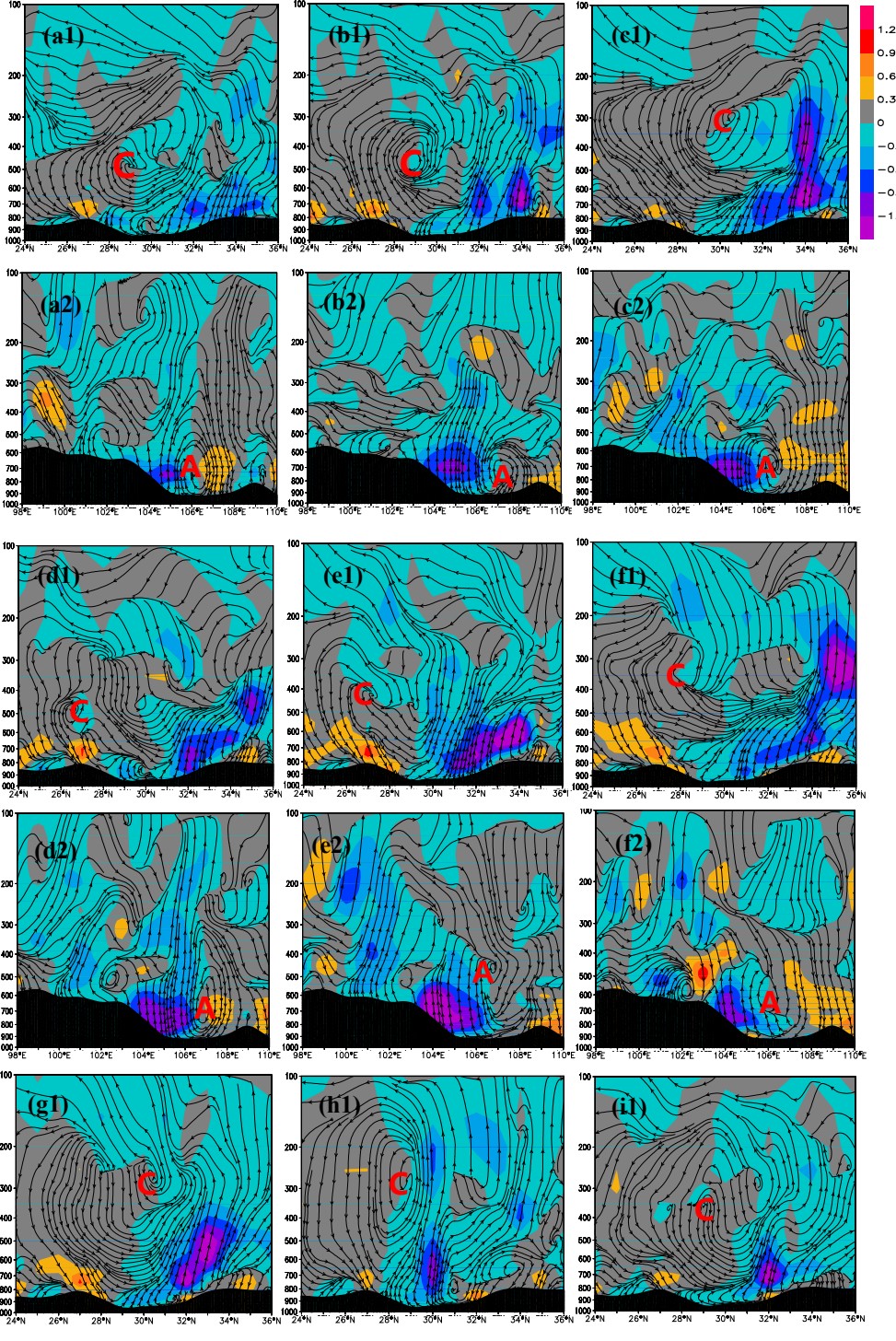

**Figure 6.** *Cont.*

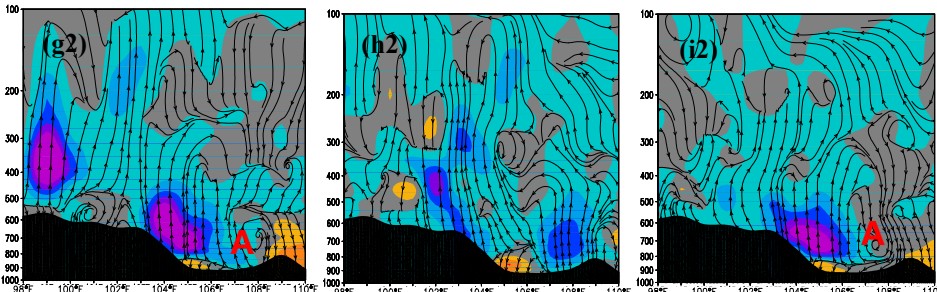

**Figure 6.** Cross section of vertical circulation along 105° E (**a1–i1**) and 32°N (**a2–i2**), in which the shaded area is the vertical velocity (Pa/s), the stream line is the synthetic circulation of both horizontal wind (m/s) and vertical velocity ($-10^{-2}$ Pa/s). A: anticyclonic vertical circulation; C: cyclonic vertical circulation. (**a1,a2**) 1200 UTC 7 July; (**b1,b2**) 1800 UTC 7 July; (**c1,c2**) 0000 UTC 8; (**d1,d2**) 1200 UTC 8 July; (**e1,e2**) 1800 UTC 8 July; (**f1,f2**) 0000 UTC 9 July; (**g1,g2**) 1800 UTC 9 July; (**h1,h2**) 0600 UTC 10 July; (**i1,i2**) 0000 UTC 11 July. Black areas: terrain.

The vertical velocity in the ascending branches of the two vertical circulations was strengthened, with its values reaching $-1.2$ Pa/s, and this led to the development of MCS$_A$ and enhanced precipitation. By 0000 UTC 8 July, the cyclonic vertical circulation had moved northward to 31° N and the vertical velocity within 32–35° N was further strengthened, resulting in the elevation of the vertical circulation center rising to 300 hPa. In the zonal direction, the ascending branch of the anticyclonic vertical circulation was still maintained over the rainstorm area, and the descending branch was located east of 106° E. This airflow structure meant that it was difficult for MCSs to develop in the descending airflow east of 106° E, while in the meridional direction the ascending airflow of 28–34° N facilitated the movement of MCSs from south to north. In the second stage, the structures of the two vertical circulations were maintained over the west of the SCB, but their intensities were different at different times. For example, at 1200 UTC 8 July, the cyclonic vertical circulation was located near 26° N and its ascending branch was below 500 hPa, while the anticyclonic vertical circulation was near 106° E and its ascending branch reached 100 hPa. The anticyclonic vertical circulation was stronger than the cyclonic circulation, and correspondingly, the negative values of vertical velocity were mainly below 600 hPa. At 1800 UTC 8 July, the cyclonic circulation strengthened and the height of its ascending branch reached 300 hPa. Meanwhile, the anticyclonic vertical circulation weakened and the height of its ascending branch was around 400 hPa. By 0000 UTC 9 July, the two vertical circulations had further weakened, but the ascending speed was still maintained in the rainstorm area. In the third stage, the height of the cyclonic vertical circulation was higher than that of the first and second stages, and was maintained, but the structural characteristics of the anticyclonic vertical circulation tended to weaken. The structure of the vertical airflow further showed that the ascending branches of the two circulations were located over the west of the SCB, which was one of the important conditional airflow structures for frequent MCS activity. Under the influence of the cyclonic vertical circulation in the meridional direction, MCSs were able to move in the south–north direction in the west of the SCB; whereas under the influence of the anticyclonic vertical circulation in the zonal circulation, it was difficult for MCSs to develop in the downdraft of the area east of 106° E. To an extent, the vertical airflow structure affected the MCS activities.

Under the influence of horizontal and vertical circulation, we further plotted the vertical helicity, divergence, and pseudo-equivalent potential temperature along the (105° E, 32° N) vertical section to characterize the dynamic and thermodynamic conditions over the rainstorm area (Figure 7). It can be seen that, regardless of latitude or longitude, positive vertical helicity and negative divergence were always maintained over the steep terrain. This shows that the precipitation weather systems developed more violently over the western part of the SCB and there was a convergence in the lower levels, which provided favorable dynamic conditions for the occurrence of heavy precipitation. The thermal conditions showed that in the zonal direction, there was a low-value center of $\theta_{se}$ (hereafter

referred to as a "low center"); plus, in the meridional direction, there were two low-value centers of $\theta_{se}$ at the mid-level (600–500 hPa) in the area south of 30° N and the low level (700 hPa to the ground) in the area north of 30° N. The maintenance of low centers caused the formation of dense $\theta_{se}$ contours over the heavy rain area, and the vertical rate of decline increased, which was beneficial to the stimulation of thermal instability. For example, at 1800 UTC 7 July, affected by the eastward expansion of $\theta_{se}$ in the zonal direction, the vertical profiles of $\theta_{se}$ over the western part of the SCB became denser and the rate of decline increased, whilst at the same time, positive vertical helicity was maintained in the southerly flow west of 106° E. In the meridional direction, the 450-hPa level at 28° N and the 700–800-hPa level over the area north of 34° N each had a low center, and correspondingly, the vertical helicity was distributed over the area south of 34° N. By 1800 UTC 8 July, the low center in the zonal direction had strengthened and the vertical profile had become denser, the vertical helicity of which below the west side was still maintained, but the convergence was strengthened, with values reaching $-7 \times 10^{-5}$/s. In the meridional direction, low centers were further developed. In particular, the low center in the low level showed a southward development trend, the vertical profiles were denser, and the decline rate increased. The vertical helicity and convergence superimposed on this low center were also strengthened, which provided conditions for the strengthening of convection and heavy rain in the second stage. By 1800 UTC 9 July, in the zonal direction, the low-center structure of $\theta_{se}$, vertical helicity, and convergence were still maintained in the heavy-rain area. In the meridional direction, the low centers in the upper level weakened but were maintained in the lower level, and the vertical helicity and convergence became deeper. By 1800 UTC 10 July, the thermodynamic characteristics were still maintained in the zonal direction, but the vertical helicity and water vapor convergence had weakened in the meridional direction, which continued from 0600 UTC 11 July to the end of the process. Clearly, the low centers of $\theta_{se}$ provided favorable thermal conditions for convection, which not only made the $\theta_{se}$ profile denser over the rainstorm area and caused thermal instability ($\partial\theta_{se}/\partial p > 0$) from the ground to 500 hPa, but was also affected by the low center of $\theta_{se}$ in the low level and its gradient increased, forming a frontal zone with an obvious vertical helicity and convergence, which provided favorable dynamic conditions for the occurrence of convection.

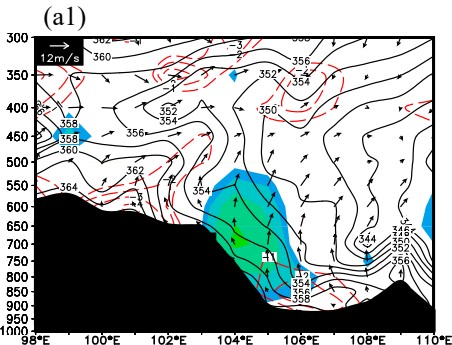
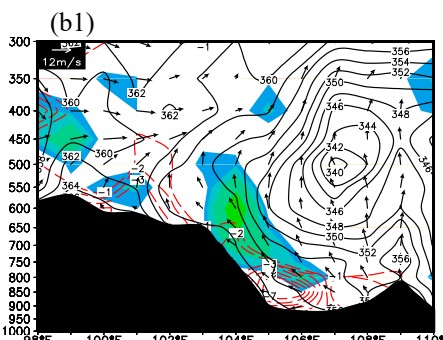

**Figure 7.** *Cont.*

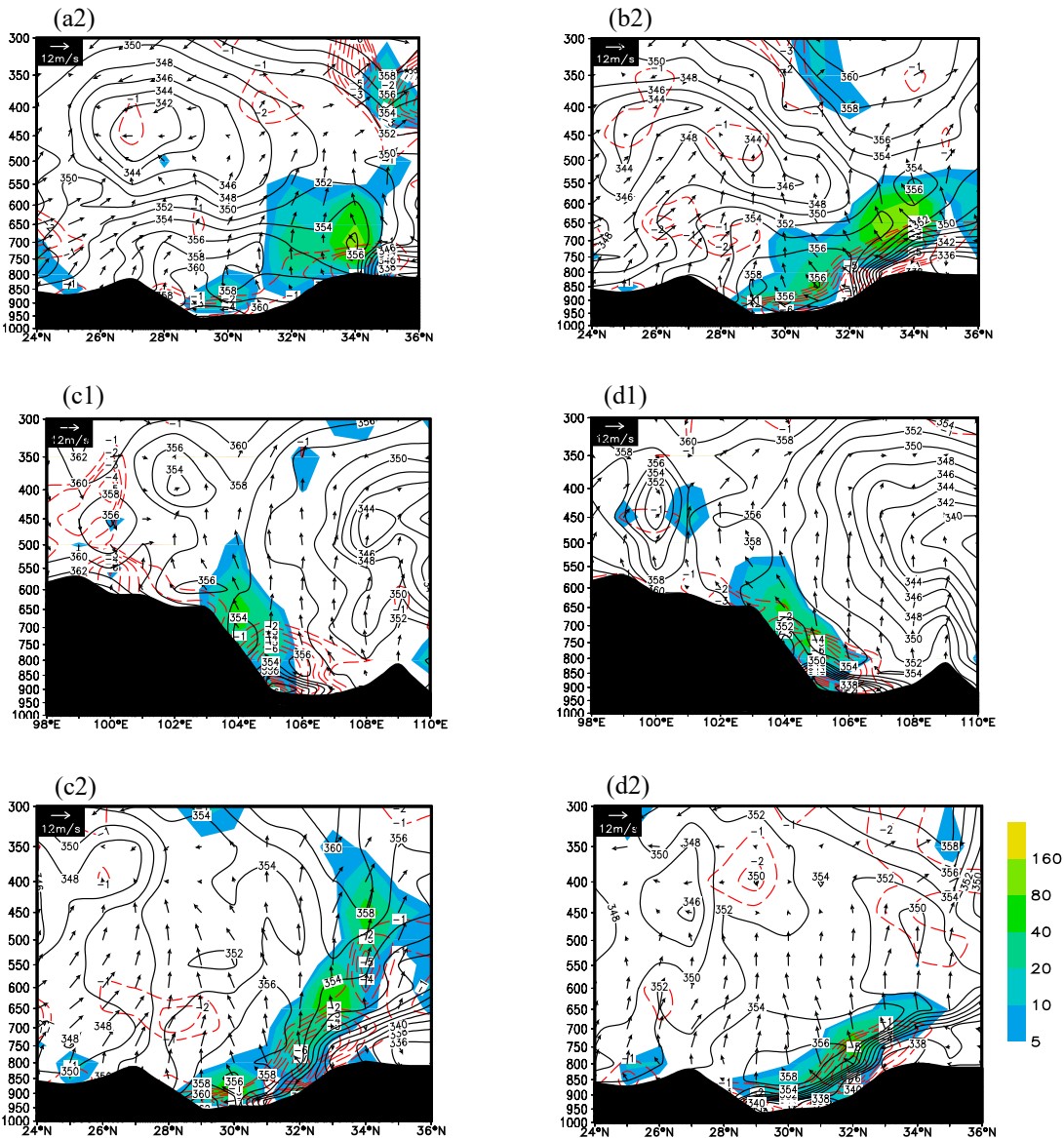

**Figure 7.** The vertical helicity (shaded, $10^{-6}$ Pa/s$^2$), pseudo-equivalent potential temperature (contours, K), negative divergence (red dashed lines, $10^{-5}$/s), and horizontal wind (m/s) distribution (**a1–d1**) vertical section along 32°N; (**a2–d2**) along 105°E; (**a1,a2**) 1800 UTC 7 July; (**b1,b2**) 1800 UTC 8 July; (**c1,c2**) 1800 UTC 9 July; (**d1,d2**) 1800 UTC 10 July. Black areas: terrain.

### 5.2. Characteristics of Convective Parameters

In order to analyze the environmental parameters of MCS activity, the NCEP_FNL data were used to draw a sounding map during the generation period of MCSs. According to Figure 3, the grid points of (30° N, 102° E) at 1800 UTC 7 July, (30° N, 103° E) at 0600 UTC 8 July, (28° N, 104° E) and (32° N, 104° E) at 1200 UTC 8 July, (31° N, 103° E) at 1800 UTC 8 July, and (31° N, 104° E) at 1800 UTC 9 July were selected to draw a sounding map to reflect the convection environment of MCS$_A$, MCS$_B$, MCS$_C$, MCS$_D$, MCS$_E$, and MCS$_F$, respectively. Meanwhile, considering that the altitude of grid point (30° N, 102° E) is about 2600 m and the near-ground layer is near 750 hPa, the grid points of (30° N, 103° E) and (31° N, 103° E) were about 700–800 m (near 925 hPa), and the grid points of (28° N, 104° E), (32° N, 104° E) and (31° N, 104° E) were about 500 m (near 950 hPa). When calculating the convection parameters, the terrain altitude was deducted, and the height of the near ground was selected as the lower limit of integration. Based on this, the main convection



parameters were calculated and the sounding map was drawn, the results of which are shown in Figure 8. It can be seen that:

(1) During the generation period of MCSs, there was a large value of convective available potential energy (CAPE) and a small value of convective inhibition (CIN). Under unstable stratification conditions, the release of CAPE is conducive to the enhancement of vertical upward movement ($\omega_{\max} = [2CAPE]^{1/2}$) [57] and causes a strong development of convective activity. In the first stage, $MCS_A$'s CAPE was 1707 J/kg and its CIN was 0/kg; and $MCS_B$'s CAPE was 546 J/kg and its CIN was 5 J/kg; in the second stage, $MCS_C$'s CAPE was 3324 J/kg, $MCS_D$'s CAPE was 4293 J/kg, $MCS_E$'s CAPE was 4225 J/kg, and the values of CAPE were further increased; in the third stage, $MCS_F$'s CAPE was 4293 J/kg, $MCS_F$'s CAPE was 2949 J/kg, and $MCS_G$'s CAPE was 209 J/kg. The variation of CAPE indicates that the unstable energy in the second stage was the largest, and corresponding MCSs developed the most strongly, which was also the strongest period of precipitation.

(2) The thermal parameters showed that the K-index [40,41] reached 39 °C in the first stage and the LI index [39] was −1 to −3 °C, indicating that the MCSs occurred in a thermally unstable environment; in the second stage, the unstable state of the atmosphere was further aggravated, the K-index was 43–47 °C, and the LI index was −7 to −9 °C; and in the third stage, the K-index reached 37–43 °C, the LI index was −6–4 °C, the unstable thermal state in the early stage was maintained, but tended to stabilize in the later stage, and the convection activity tended to weaken.

(3) The T-log P charts in the generation period of the MCSs showed that there were vertical wind shears in the middle to lower levels, for instance, $MCS_A$, $MCS_B$, $MCS_C$, and $MCS_F$ showed there were vertical wind shears within 850–700 hPa; and $MCS_D$, $MCS_G$, and $MCS_H$ had vertical wind shears within 500–600 hPa.

(4) The water vapor parameters showed that the precipitable water (PW) was maintained between 3.5 and 5.7 cm in the first stage and the water vapor was close to saturation from 600 hPa to the ground, but was relatively dry at 600–250 hPa, which formed the "upper dry–lower wet" structure; in the second stage, PW was maintained between 6.5 and 7.6 cm and continued to increase. Except for $MCS_C$, the humidity was close to saturation from 200 hPa to the ground, which was a deep and wet convection environment. The high PW and humidity conditions provided a favorable water vapor environment for MCSs to produce extreme rainstorms. Therefore, MCSs were developed in an environment with high CAPE, high humidity, and unstable stratification. In the second stage, in particular, unstable energy and water vapor were increased and the thermal state was very unstable, which led to strong development of MCSs and the occurrence of extreme rainstorms.

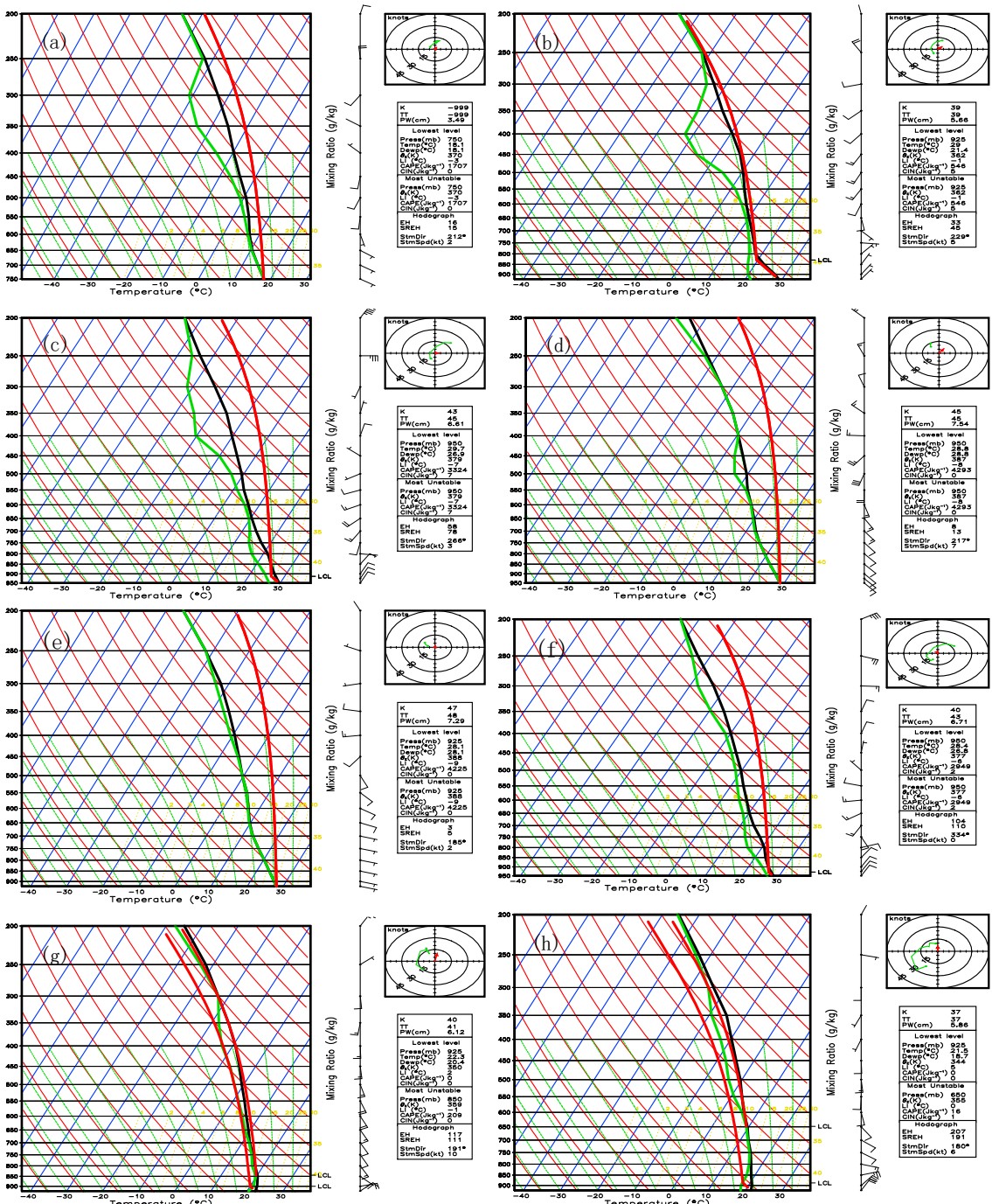

**Figure 8.** T-Log P chart analysis in the period of MCS generation. In (**a–h**), the thick red line is stratification temperature, the thick black line is environmental temperature, the thick green line is the dew point, the thin blue line is the temperature line, the green dotted line is the wet insulation line, and the red thin line is the dry adiabatic line. (**a**) MCS$_A$ T-Log P of (30° N, 102° E) at 1800 UTC 7 July; (**b**) MCS$_B$ T-Log P of (30° N, 103° E) at 0600 UTC 8 July; (**c**) MCS$_C$ T-Log P of (28° N, 104° E) at 1200 UTC 8 July; (**d**) MCS$_D$ T-Log P of (32° N, 104° E) at 1200 UTC 8 July; (**e**) MCS$_E$ T-Log P of (31° N, 103° E) at 1800 UTC 8 July; (**f**) MCS$_F$ T-Log P of (28° N, 104° E) at 1200 UTC 9 July; (**g**) MCS$_G$ T-Log P of (31° N, 104° E) at 1800 UTC 9 July; (**h**) MCS$_G$ T-Log P of (31° N, 104° E) at 0000 UTC 10 July.

In addition, from the observed result of Wenjiang Sounding Station, which was located near the area of the rainstorm center (Table 1), the following can be concluded:

(1) CAPE had a large value in the first and second stages, but it was basically close to zero in the third stage. At 1200 UTC 7 July with the rainstorm approaching, the CAPE was beyond 2000 J/kg, and by 0000 UTC 8 July the value had dropped to 700 J/kg. In this period, the convective unstable energy was released and MCS$_A$ developed strongly. In the second stage, the convection conditions were established again, the CAPE value reached 1600 J/kg at 1200 UTC 8 July, and by 0000 UTC 9 July, the CAPE had decreased significantly. In this process, MCS$_C$, MCS$_D$, and MCS$_E$ developed strongly in the rainstorm area. The observed data also indicated that a larger CAPE value was one of the important conditions for MCS activities, with strong activities of MCSs generally occurring during the release of CAPE.

(2) In the first and second stages, the K- and LI indexes were 39–40 °C and −1 °C to −3 °C, respectively, and the surface water vapor was 19–22 g/kg, indicating that the atmosphere over the rainstorm area was unstable and close to saturation. In the third stage, the values of convection parameters decreased (e.g., the K- and LI indexes were 31–36 °C and 1–3 °C, respectively), and the surface water vapor was 16–19 g/kg, indicating that the atmosphere still maintained a high water vapor content and was close to saturation. From the variation of the LFC and CCL, the levels in the first and second stages were generally higher than those in the third stage, which further demonstrates that the convection activities of the first two stages were stronger than in the third stage.

**Table 1.** Physical value variation of MCSs based on sounding-station data: CAPE, CIN, KI, LI, level of free convection (LFC), convection condensation level (CCL), surface vapor mixing ratio ($Q_{surf}$), and relative humidity (Rh).

| | 1200 UTC 7 July | 0000 UTC 8 July | 1200 UTC 8 July | 0000 UTC 9 July | 1200 UTC 9 July | 0000 UTC 10 July | 1200 UTC 10 July | 0000 UTC 11 July |
|---|---|---|---|---|---|---|---|---|
| CAPE (J/kg) | 2164 | 770.6 | 1634.8 | 128.7 | 0 | 0 | 5.5 | 0 |
| KI (°C) | 40 | 39 | 40 | 39 | 31 | 36 | 31 | 33 |
| LI (°C) | −2.57 | −0.42 | −2.02 | −1.03 | 1.41 | 1.91 | 3.52 | 2.63 |
| LFC (hPa) | 832 | 816.4 | 855.4 | 700.4 | —— | —— | 897 | —— |
| CCL (hPa) | 858 | 857 | 891 | 890 | 915 | 871 | 939 | 941 |
| $Q_{surf}$ (g/kg) | 21.34 | 20.08 | 21.41 | 18.94 | 18.94 | 16.7 | 16.65 | 16.62 |
| Rh (%) | 81–84 | 80–94 | 80–94 | 88–94 | 77–100 | 88–94 | 82 | 88–100 |

### 5.3. Influence of Terrain

Past studies have shown that gravity waves have a considerable influence on the convective activities of rainstorms [58–60]. Leeward slopes of mountain ranges, geostrophic adaptation, and terrain forcing are important mechanisms and sources of wave energy for large-amplitude mesoscale gravity waves, and this is particularly the case for undulating terrain, where the topographical gravity waves interact with the proper airflow, which often causes heavy or increased precipitation. Due to the undulating terrain in the west and north of the SCB (altitude is 500–3000 m in the west and 1000–2500 m in the north), when warm–moist airflow in the low-level interacts with the terrain, it often causes heavy precipitation. In order to analyze the relationship between the terrain gravity wave disturbance in the west and north of the SCB and the low-level airflow, we analyzed the profile of gravity wave stress, the 700-hPa horizontal wind field, and vertical velocity along 104° E and 32° N (Figure 9).

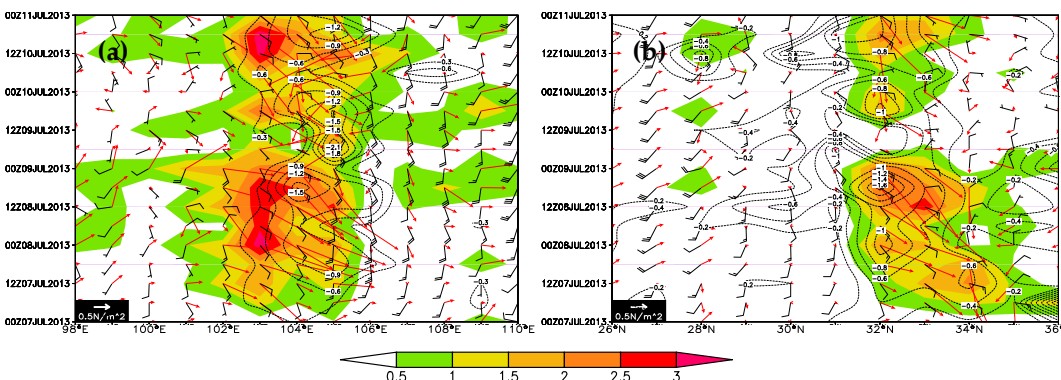

**Figure 9.** Temporal evolution of the surface gravity wave stress (red arrow and filled area, N/m$^2$), 700-hPa vertical velocity (dotted lines, Pa/s) and wind field (black, m/s) along 32° N (**a**) and 104° E (**b**) based on NCEP_FNL data.

In this figure, the red arrow denotes the surface gravity wave stress vector, the color fill indicates the magnitude of the gravity wave stress, the dashed line is the 700-hPa vertical velocity, and the black wind vector is the 700-hPa wind field. The zonal evolution shows that the area near 102–105° E of the terrain leeward slope was an area with large gravity wave stress values, and its direction was southeast. Under the action of terrain gravity waves, a blocking and lifting effect was formed on the easterly wind, which was beneficial to horizontal convergence and upward movement, and correspondingly, there was a feature of ascending motion between 103° E and 105° E at 700 hPa. The meridional direction also showed similar characteristics; under the action of the Daba Mountain terrain, 32–34° N was a large-value area of gravity wave stress and its direction was east north, which formed a blocking effect on the northeast wind. This shows that, during the transportation of low-altitude warm–moist airflow to the rainstorm area, it was affected by the drag of terrain gravity waves.

Figure 10 shows the surface gravity wave stress (red vectors, N/m$^2$), 700-hPa vertical velocity (contours, $10^{-1}$ Pa/s), wind field (wind poles, m/s), and its negative divergence (shaded, $10^{-5}$ s$^{-1}$). Obviously, the leeward east slope (A) and the windward south slope (B) of the plateau terrain are both areas with large gravity wave stress; and under its influence, on one hand, the warm–moist southwestern airflow in the low-level rose up on the south windward slope and caused violent upward movement; whilst on the other hand, the southwestern warm–moist airflow turned to form southeast wind over the southern part of the basin, and formed strong upward movement in the western part of the basin, which provided sufficient dynamic conditions for the occurrence of the rainstorm. In the first stage (Figure 10a–d), the gravity wave stress along the mountain in the western part of the basin reached 2–5 N/m$^2$, the 700-hPa southwest wind changed to southerly wind, and an obvious upward movement (values up to $-8 \times 10^{-1}$ Pa/s) was built along the mountain terrain in the western part of the basin. However, from the perspective of the convergence of the 700-hPa wind field, the negative divergence range was small and mainly located along the mountain in the western part of the basin, and correspondingly, the range of the rainstorm at this stage was also smaller. This shows that strong precipitation basically occurred in the areas with large values of gravity wave stress and obvious vertical upward movement. In the second stage (Figure 10e–h), the mountains along the western part of the SCB and the southeast plateau were areas with obvious gravity wave stress and values up to 2–6 N/m$^2$. At 700 hPa, there was a southeast wind flow over the western part of the basin with wind speeds of 8–12 m/s and its direction was the opposite of the gravity wave stress. This configuration was beneficial to make the terrain gravity waves drag the low-level southeast wind and be uplifted along the mountain in the western part of the basin, forming a more violent vertical upward movement. The vertical velocity center value reached $10–20 \times 10^{-1}$ Pa/s. In addition, the 700-hPa convergence was further strengthened with values of $-2$ to $-6 \times 10^{-5}$/s, which provided sufficient water vapor conditions for

the intensification of rainstorms. From the statistical observation of precipitation, this stage was indeed stronger than the first stage. For the third stage (Figure 10i–l), the action of topographic gravity waves and the configuration of the low-level wind field were basically the same as in the previous two stages, and resulted in heavy precipitation.

Clearly, the gravity waves near the terrain play an important role in heavy rain occurring along the mountains in the western part of the SCB. In the mountainous area of the western part of the SCB, the gravity wave stress was obvious and its direction was the opposite to the direction of the lower southeast warm–moist airflow. This configuration can form a drag effect in the low-level airflow, which was conducive to the convergence of the wind field and the strengthening of the vertical ascending motion, and then further conducive to the development of the convective system and the enhancement of precipitation. During the entire rainstorm, obvious vertical upward movement was maintained in the area where the terrain and the southeast wind acted. Particularly in the period when the convergence of the horizontal wind field was obvious, the resultant precipitation was more severe.

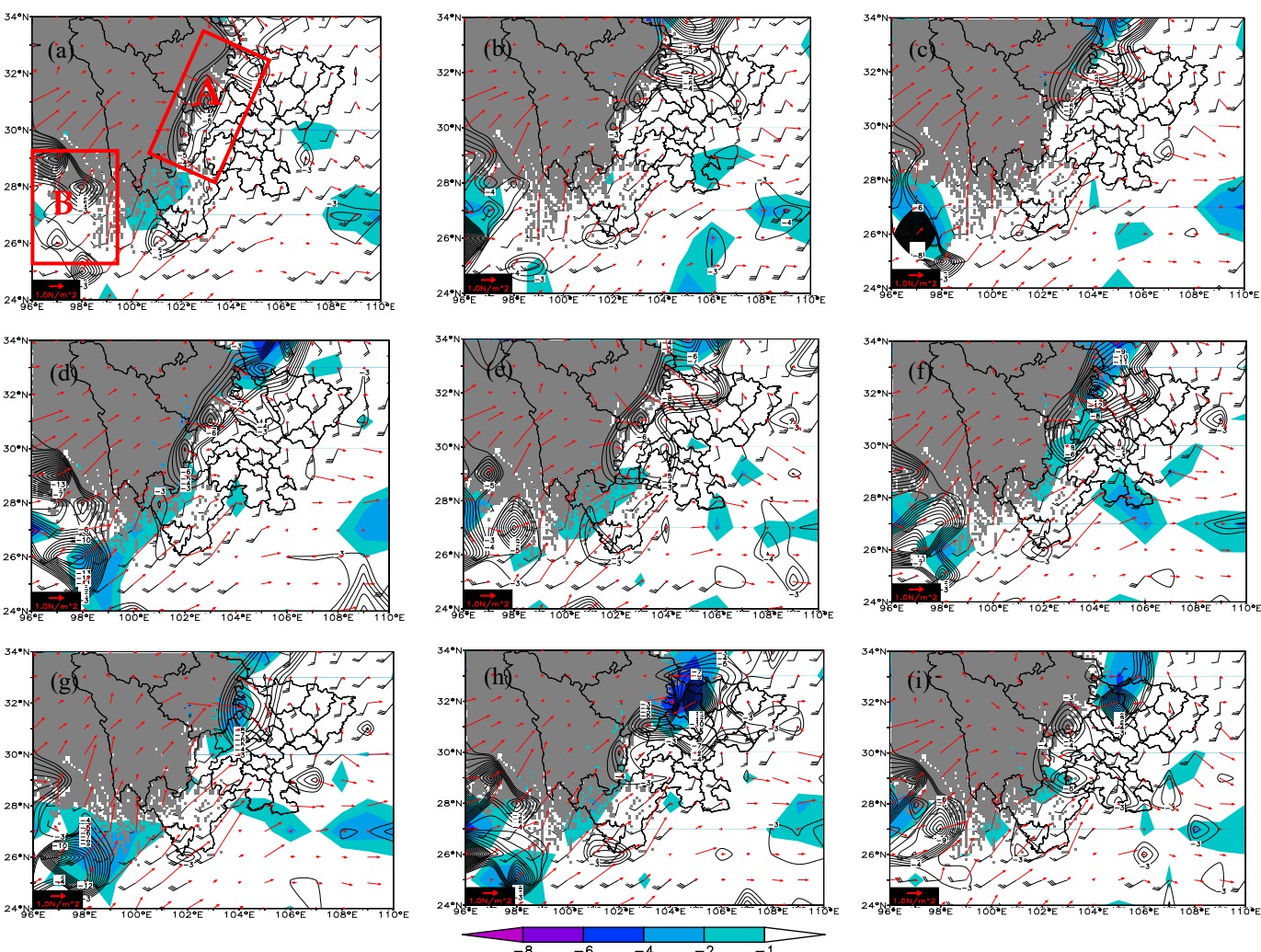

**Figure 10.** *Cont.*

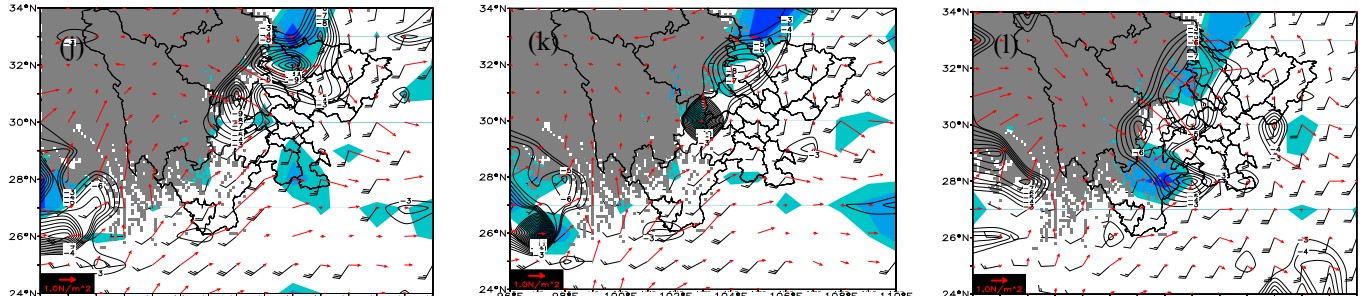

**Figure 10.** As in Figure 9, but for the gravity wave stress vector (red, N/m$^2$), 700-hPa vertical velocity (contours, $10^{-1}$ Pa/s), wind field (wind poles, m/s) and negative divergence (color fill, $10^{-5}$/s): (**a**) 1200 UTC 7 July; (**b**) 1800 UTC 7 July; (**c**) 0000 UTC 8 July; (**d**) 0600 UTC 8 July; (**e**) 1200 UTC 8 July; (**f**) 1800 UTC 8 July; (**g**) 0000 UTC 9 July; (**h**) 0600 UTC 9 July; (**i**) 1200 UTC 9 July; (**j**) 1800 UTC 9 July; (**k**) 0000 UTC 10 July; (**l**) 1200 UTC 10 July. Grey areas: terrain above 700 hPa.

## 6. Conclusions

Against a stable large-scale circulation background, the rainstorm investigated in the present study was caused by MCSs. The following conclusions regarding this extreme heavy rainstorm can be drawn:

(1) The continuous activity of MCSs was a direct cause of the formation of extreme rainstorms. Under an "east high and west low" circulation mode, MCSs were active in the trough and shear zone. The MCSs over the plateau area weakened at night, while they strengthened over the basin at night, which formed a "cloud cluster wave train" phenomenon from the plateau to the basin.

(2) MCSs over the plateau and basin were active in an environment with large CAPE values, high humidity, and unstable stratification. However, compared to MCSs over the plateau, the development of MCSs over the basin was also related to the following conditions: (a) The activities of MCSs over the rainstorm area were related to ascending branches of two vertical circulations. Under the influence of meridional vertical circulation, MCSs could move toward the south–north direction of the western basin; and under the influence of zonal circulation, it was difficult for MCSs to develop in the descending airflow east of 106° E. To an extent, the vertical airflow structure affected the direction of MCS activity. (b) Thermodynamic conditions showed that, regardless of latitude or longitude, positive vertical helicity and negative divergence were always maintained over the steep terrain, and a low-value center of $\theta_{se}$ in the zonal direction and two low-value centers of $\theta_{se}$ in the meridional direction were also established. This not only made the $\theta_{se}$ profile denser over the rainstorm area and caused thermal instability ($\partial\theta se/\partial p > 0$) from the ground to near 500 hPa, but was also affected by the low-center of $\theta_{se}$ in the low level. The gradient of $\theta_{se}$ increased and formed a frontal zone with an obvious vertical helicity and convergence, and this provided favorable dynamic conditions for the occurrence of convection. (c) The development of MCSs was also related to topographic gravity waves, particularly in the mountainous area of the western part of the SCB where the gravity wave stress was obvious and its direction was the opposite to the direction of the lower southeast warm–moist airflow. This configuration can form a drag effect in the low-level airflow, which was conducive to the convergence of the wind field and the strengthening of the vertical ascending motion, and then further conducive to the development of the convective system and the enhancement of precipitation. These findings will help us understand the similarities and differences between the convection conditions over the plateau and the basin, particularly about the evolution of MCSs in the formation of extreme rainstorms under complex terrain and the relationship between vertical circulation and topography.

(3) It is important to note that as a typical area of China's rainstorm weather, the Sichuan region is known within the country for its high level of incidence regarding torrential

rain and flooding. The observational results of stationary meteorological satellite showed that many rainstorm events in the region are related to MCSs activities, although only an extreme rainstorm was selected in this paper, the activity of MCSs was representative. Normally during rainstorms, MCS activities can be observed over both the plateau and the basin. However, in fact, the precipitation efficiency of MCSs in the basin is higher than that of the plateau, which may be related to the altitude, water vapor, and low-level system of the plateau. This difference needs to be considered when we analyze MCSs in actual forecasts. Rainstorms induced by MCSs mainly occur in the basin.

**Author Contributions:** Y.C.: Conceptualization, formal analysis, methodology, software, validation, data curation, writing—original draft preparation. Y.L.: Conceptualization, methodology, writing—review and editing, funding acquisition. All authors have read and agreed to the published version of the manuscript.

**Funding:** This research was founded by a Strategic Priority Research Program of the Chinese Academy of Sciences (No. XDA23090103); a project of Science & Technology Department of Sichuan Province (2021YFS0326); a special project of the forecaster of China Meteorological Administion (CMAYBY2020-113), and the National Natural Science Foundation of China (No. 91937301).

**Institutional Review Board Statement:** Not applicable.

**Informed Consent Statement:** Not applicable.

**Data Availability Statement:** All the data used in this study are as follows: (1) sounding observational data, and land surface precipitation data are from observational systems of China Meteorological Administration. The datasets generated during and/or analyzed during the current study are available from the corresponding author on reasonable request. (2) Global final (FNL) analysis data from the National Centers for Environmental Prediction (NCEP), with a spatial resolution of $1° \times 1°$ and temporal resolution of 6 h (hereafter referred to as NCEP_fnl data); the data can be freely downloaded from the link http://rda.ucar.edu/ (accessed on 12 March 2021). (3) Brightness temperature (Tb) data from the FY2D geostationary weather satellite provided by the National Satellite Meteorological Center (NSMC), China Meteorological Administration (http://www.nsmc.org.cn/en/NSMC/Home/Index.html (accessed on 12 March 2021)).

**Acknowledgments:** We sincerely thank the anonymous experts for their valuable comments on this paper.

**Conflicts of Interest:** The authors declare no conflict of interest.

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
