# Peer review of "Convective Characteristics and Formation Conditions in an Extreme Rainstorm on the Eastern Edge of the Tibetan Plateau"

_atmosphere, doi:10.3390/atmos12030381_

Round 1

Reviewer 1 Report

Attached

Author Response

Dear reviewer,

Thank you so much for taking the time to read my cover letter. According to your comment, I revised the manuscript (ID: 1127399) again. I revised and responded to the main issues, please see the attachment.

Reviewer 2 Report

The paper titled "Convective characteristics and formation conditions in an extreme rainstorm on the eastern edge of the Tibetan Plateau" raises interesting issues related to rainstorm events associated with mesoscale convection systems (MCSs).

Despite the seriousness of the issues raised, the article was written very carelessly. The structure of the article should be changed. First of all, the authors should pay attention to the description of all applied research methods in the section describing the used research methods. After the description of research methods, only the obtained research results should be described in the following chapters. Also, the article is too long. Authors should make appropriate abbreviations of the submitted content. The conclusions at the end of the article are too short and refer to only some of the research results. The article is not suitable for printing in its current form.

Author Response

Dear reviewer,

Thank you so much for taking the time to read my cover letter. According to your comment, we revised the manuscript (ID: 1127399) again. I revised and responded to the main issues as follows:

Despite the seriousness of the issues raised, the article was written very carelessly. The structure of the article should be changed. First of all, the authors should pay attention to the description of all applied research methods in the section describing the used research methods. After the description of research methods, only the obtained research results should be described in the following chapters. Also, the article is too long. Authors should make appropriate abbreviations of the submitted content. The conclusions at the end of the article are too short and refer to only some of the research results. The article is not suitable for printing in its current form.

  1. the article was written very carelessly.

Dear reviewer, thank you very much for your valuable comments. In response to this extreme event, we believe that the research is helpful to understand the evolution of MCSs in the formation of extreme rainstorms under complex terrain and the relationship between vertical circulation and topography. The topic of this paper is representative.

Indeed, there are some problems in the article. Despite this, we strive to make the quality of the manuscript reach the level of publication in accordance with expert comments and repeated revisions by ourselves. Of particular after anonymous experts’ comments and our revisions, the quality of the article has further improved this time.

2. The structure of the article should be changed. First of all, the authors should pay attention to the description of all applied research methods in the section describing the used research methods. After the description of research methods, only the obtained research results should be described in the following chapters.

 Dear reviewer, your suggestion is also a good idea. As far as the current structure, we think it is also feasible.  Following this introduction, section 2 describes the data sources and methods; section 3 analyzes the observed precipitation and the systems that directly induced it; section 4 analyzes the evolution of clouds in the MCSs; section 5 explains the conditions that were favorable for the MCS activities; and section 6 concludes the study.

3. Also, the article is too long. Authors should make appropriate abbreviations of the submitted content.

Due to we only select an extreme rainstorm event, which occurred in the representative complex terrain area near the eastside of Tibetan plateau. In order to study the convective evolution and the change of dynamic and thermal conditions in a more detailed way, so we provide a lot of plots to support our conclusion. Overall, the article is very long but we think it is necessary.        

4. The conclusions at the end of the article are too short and refer to only some of the

research results. The article is not suitable for printing in its current form.

Dear reviewer, I rewrote conclusion (3) and added an appropriate sentence at the end of conclusion (2), see lines 656-671

Reviewer 3 Report

The authors present a detailed meteorological analysis of an extreme precipitation event with more than 1000mm rain within 4 days in the Sichuan province of China. To understand the event, the authors combined a wide range of available information including rain gauge data, satellite derived cloud top temperatures, synoptic scale information and reanalysis data to calculate CAPE, CIN and further convection parameters as well as helicity.

While the authors describe explicitly how they calculate pseudo-equivalent potential temperature and how helicity is defined, they use CAPE, CIN, level of free convection, lifted index and K index without introducing them. I think it would be too much to write down their definitions, but it would be good to cite papers which introduce these parameters to help those readers who are not familiar with them.

There are many small plots with sometimes very small text in the manuscript. This is something this reviewer does not like. Especially Fig. 8 contains some unused information: The hodographs and further information like storm-relative helicity are not discussed in the paper. Maybe the authors should consider to leave them out and create more space for the information they discuss.

This reviewer is impressed by the wide range of information used in this paper. The meteorological analysis goes far beyond what is usually done to analyze the reasons for extreme precipitation events. It shows how orography and meteorological processes on various spatial scales act together. While the results are specific for one region in China, this reviewer hopes that this paper gives a good example to those who analyse extreme meteorological events in other regions of the world.

I recommend the manuscript for publication with minor (mainly grammatical) changes.

Minor Points

L90: ‘evolution of clouds’, not ‘cloud’

L113: Please replace ‘rotation, and its expression (Lilly 113 1986; Lu and Gao 2003) is defined as’ by ‘rotation. It is defined as (Lilly 113 1986; Lu and Gao 2003)’, since you do not define an expression but a variable.

L115: please add the V is the windspeed vector, replace ‘is three-dimensional vorticity’ by ‘ is the three-dimensional …’, replace xyz-helicity by xyx component of helicity

L119: It would be nice if the sentence starting in this line could be supported by a citation

L123: ‘In addition, due to the rainstorm occurred near the terrain, the effect of surface gravity waves stress was further discussed.’ This sentence might fit into a summary. Actually, gravity waves were not discussed at this point. It is not mentioned so far that the rainstorm occurred near the terrain, nor what that actually means.

L124: replace ‘According to references (Xu et al.2012,…’ by ‘According to Xu et al.2012,…’, i.e. cite in text instead of parentheses.

L128ff: use the same \phi as in eq. (4).

L133: systes?

L134: ‘This defining extreme precipitation usually adopts the threshold method—that is, when the precipitation value exceeds the threshold of a certain percentile in the total samples, the precipitation can be called an extreme precipitation.’ Should be ‘Extreme precipitation can be defined by the threshold method—that is, when precipitation exceeds a threshold, which can be a certain percentile of the total samples, the precipitation is called an extreme.’

L165: ‘elevation’ seems better than ‘terrain’ since elevation is the terrain feature shown and actually has the unit meter.

Fig 1b: Legend text too small to read. Also axis labels could be larger.

Fig 1d: I wouldn’t use the phrase ‘station number’ which I would associate with the number (identifier) of a station. I’d rather say ‘number of stations exceeding 50mm/h’. I’d also prefer different colours instead of a mixture of line and bar plot.

L174: ‘rainstorm stations’? I guess it’s AWS. I’d call them rain gauges.

L181: replace ‘h,’ by ‘hours,’

L184: ‘extreme rainstorm process.’ to ‘extreme rainstorm event.’

L191: ‘weakened’ or ‘weaker’?

L199: ‘In Fig.d,’ to ‘In panel d,’ (it’s all in Fig. 2.)

L244: ‘was also big.’ to ‘was also high.’ (there can be high precipitation, not big precipitation)

L249: ‘Tb showed MCSs of Tb < −50℃ were situated’ to ‘Tb showed MCSs of Tb < −50℃ situated’

L299: ‘moved over the SCB’ to ‘moved over the SCB’

L305: ‘as the first stage.’ to ‘as during the first stage.’

L477: ‘unstable energy needed for convection was further increased’ What is unstable energy? As far as I see it, instability increased.

L482ff: Neither the K index nor LI are formally introduced. Please cite them. Note that LI index means index index since the I in LI stands for index.

L489ff: refer to Fig. 8 and replace ‘T-log P maps’ by ‘T-log P charts’.

L527: ‘Physical value analysis of MCSs’. I wouldn’t call the observations an analysis.

L558: ‘dotted line,’ to ‘dotted lines,’

Fig. 10: I’d say the area indicating high elevation is grey. And what are the two rectangles?

L657: ‘This difference need to be distinguished , when we analyze MCSs in actual forecast. Rainstorms induced by MCSs mainly occur basin.’ to ‘This difference needs to be considered, when we analyze MCSs in actual forecasts. Rainstorms induced by MCSs mainly occur in the basin.’

Author Response

Dear reviewer,

Thank you so much for taking the time to read my cover letter again. According to your comment, we revised the manuscript (ID: 1127399) again. I revised and responded to the main issues , see the attachment.

Reviewer 4 Report

This is a quite interesting paper dealing with the convective characteristics and formation conditions in an extreme rainstorm event. The manuscript is well written and structured. Although it is not characterized by novelty, the analysis is interesting and can be helpful in the scientific community.

The only comment that should be made at this point is the quality of the figures. Here are some examples:

  • Figure 1b should have a better quality and larger numbers/letters.
  • Figures 2d and 2e have different sizes in the y-axis although they represent the same.
  • In Figure 3, several plots are misplaced.
  • In figure 9 the letter (a) is half-hidden.

In general the figures need to be double-checked. As far as I understand the authors use tables to place the different figures. They should consider merging them using some software and placing them within the manuscript as a single image to avoid misplacement.

Author Response

Dear reviewer,

Thank you so much for taking the time to read my cover letter. According to your comment, we revised the manuscript (ID: 1127399). I revised and responded to the main issues.

Dear reviewer,

Thank you so much for taking the time to read my cover letter. According to your comment, we revised the manuscript (ID: 1127399). I revised and responded to the main issues as follows:

1. Figure 1b should have a better quality and larger numbers/letters.

I have modified the Figure 1b, see between line 168 and line 169 in the revised version.

2.Figures 2d and 2e have different sizes in the y-axis although they represent the same.

Dear reviewer, I have further adjusted the size of the two panels and kept them as consistent as possible, see between line 198 and line 199 in the revised version

3. In Figure 3, several plots are misplaced.

I have further checked and adjusted positions of some plots and kept them as consistent as possible, see between line 263 and line 264 in the revised version

4.In figure 9 the letter (a) is half-hidden.

I have modified it, see line 557 and line 558 in the revised version

5. In general the figures need to be double-checked. As far as I understand the authors use tables to place the different figures. They should consider merging them using some software and placing them within the manuscript as a single image to avoid misplacement.

Dear, I carefully checked all the Figures again, Due to t the manuscript was revised on the basis of the first submitted manuscript, in which the Figures of the manuscript are placed in the tables and has been initially edited by the editor. If the manuscript is accepted for publication, these tables will be cancelled and a printed version will be formed.

Round 2

Reviewer 2 Report

The article has been partially corrected in accordance with the comments of the reviewer. There is no description of the convection parameters and PW as well as the T-Log P chart analysis in the research methods chapter. After filling the gaps, in my opinion, the article is ready for printing.

Author Response

Dear reviewer,

According to your comment, we revised the manuscript (ID: 1127399) again. I revised and responded to the main issues as follows:

The article has been partially corrected in accordance with the comments of the reviewer. There is no description of the convection parameters and PW as well as the T-Log P chart analysis in the research methods chapter. After filling the gaps, in my opinion, the article is ready for printing.

I have added the description and calculation formula of the convection parameters involved in the paper, see lines 103-127, and I also added 3 citations (the 11th, 17th, 19th in section of references) in the revised version.

Since the abbreviations of convection parameters first time appeared in this section” data and methods”, so only the abbreviations are used in the title of in Table 1. And I modified it, see lines 556-558 in the revised version.

The T-Log P chart is an important map for convection, which includs important Thermal and dynamic process, water vapor variation in vertical direction, potential convective instability energy. In the T-Log P chart, there are three important lines in the vertical profile change, as described in the Fig.8, the thick red line is stratification temperature, the thick black line is environmental temperature, and the thick green line is the dew point. I think the title of Fig.8 about the introduction of t-log p chart is clear and the values of convective parameters are specific. 

This manuscript is a resubmission of an earlier submission. The following is a list of the peer review reports and author responses from that submission.

Round 1

Reviewer 1 Report

The detailed analysis of atmospheric processes during heavy rainfall over Sichuan Basin in 7-11 July 2013 is presented. The rainstorms in the region often cause significant damages, so the study is of great importance. The study was well conducted using combination of instrumental data: sounding radio data, satellite remote sensing data and NCEP FNL Operational Global Analysis data. The processes in each stage of rainstorm were analyzed comprehensively. The formation of subsequent mesoscale convective systems as the main cause of all three stages of heavy rainfall was resulted from the specific circulation mode. The detailed analysis of vertical circulation, convection and the placement of large values of gravity waves stress resulting development of mesoscale convective systems was presented.

The results will be of interest to readers. The manuscript corresponds to the Journal scope and is recommended for publication after correction of few typos.

Notes:

Line 79: “form” – maybe “from”?

Figure 3: The charts (c5)–c(16) are missing?

Figure 5: the caption is missing.

Figure 9, caption: “along 104°E (a) and 32°N (b)” – maybe “along 32°N (a) and 104°E (b)”?

Reviewer 2 Report

Reluctantly, I recommend rejection of this paper.  The investigation and documentation have merit, and the subject is of interest to a broader readership, but the manuscript itself has a number of flaws which I urge the authors to address before any resubmission.

The topic of heavy convective rainfall is certainly of interest, and the case study documents an interesting and important example. It seems to me, however, that the manuscript was prepared in haste, without adequate proofreading. In particular:

several panels from some figures are missing (in Figure 1, panel b is repeated and there is no panel c; in Figure 3 panels c5 through c16 are missing); 

the caption to Figure 1 is very confusing - at least partly because of the missing panel, and the caption to Figure 5 is entirely missing; 

sections of the text are partly repeated, confusing this reviewer at least (e.g.lines 339-349);

discussion of convective parameters need to be linked to climatological values, to allow readers to identify where those parameters have reached locally extreme values e.g. Section 5.2; 

the expression in some places makes it difficult to understand what the authors intended e.g. lines 575-578. I think I understand what the authors mean here, but am not sure. There are several other examples.

several references are missing, which is a pity because the references in general seem relevant and topical.

There are a number of other more minor errors in the text, but I don't detail them in light of the substantial changes required to bring the manuscript up to an acceptable standard for publication. Rather, I urge the authors to carefully proofread the document prior to any resubmission, and ideally to request an independent colleague to carefully proofread for them. I understand that manuscripts can contain errors that authors simply miss because of close familiarity, but missing whole panels or captions of figures goes beyond that. Still, as above, I suggest the authors resubmit a carefully proofread manuscript in the future, as much of the content is of broad interest to the meteorological community. 

Reviewer 3 Report

See attached
